# Bayesian-LoRA: Probabilistic Low-Rank Adaptation of Large Language Models

**Moule Lin** [1]  **Shuhao Guan** [2]  **Andrea Patane** [1]  **David Gregg** [1]  **Goetz Botterweck** [1]

## Abstract

Large language models are typically optimized for accuracy and, therefore, will guess even when uncertain about their predictions. This problem becomes especially pronounced when the model is fine-tuned on small datasets, which often causes overfitting and results in a tendency toward miscalibration. In this work, we introduce Bayesian-LoRA, which reformulates the deterministic LoRA update as a probabilistic low-rank representation inspired by Sparse Gaussian Processes (SGP). We identify a structural isomorphism between LoRA's factorization and Kronecker-factored SGP posteriors, and show that LoRA emerges as a limiting case when posterior uncertainty collapses. We conduct extensive experiments on various LLM architectures across commonsense reasoning, language modeling, and mathematical reasoning benchmarks. With only approximately 0.42M additional parameters and $\approx 1.2\times$ training cost relative to standard LoRA, Bayesian-LoRA significantly improves calibration across models from 7B up to 30B, achieving up to 84% Expected Calibration Error (ECE) and 76% Negative Log-Likelihood (NLL) reduction while maintaining competitive accuracy for both in-distribution and out-of-distribution (OoD) evaluations.

**Code**: https://github.com/moulelin/Bayesian-LoRA

## 1. Introduction

Fine-tuning large language models (LLMs) with parameter-efficient methods such as Low-Rank Adaptation (LoRA) (Hu et al., 2022) is now standard practice for adapting pre-trained models to downstream tasks (Ding et al., 2023; Xu et al., 2021; Malladi et al., 2023; Hu et al., 2024; Wang et al., 2024a; Shorinwa et al., 2025; Li et al., 2025). By updating only a small set of low-rank matrices, LoRA achieves strong task accuracy with lower computational cost, memory footprint, and data requirements (Yin et al., 2024; Lin et al., 2024; Liu et al., 2025; Ye et al., 2024).

Although pre-trained models are reasonably well calibrated out of the box (Zhou et al., 2024; Wang, 2023), calibration often deteriorates after domain-specific fine-tuning, and as a result, models tend to become systematically overconfident (Liu et al., 2024; Mai et al., 2024; Zhou et al., 2023). Such degradation poses problems when LLMs are applied in safety-critical domains, such as autonomous driving (Tu et al., 2025; Wu et al., 2021), medical diagnosis (Savage et al., 2025), and others (Liu et al., 2025; Kim et al., 2025; Chuang et al., 2025). This motivates calibration-aware fine-tuning methods that can maintain or improve probability calibration during LLM training.

**Limitations of existing approaches.** Probabilistic methods can improve calibration through Bayesian neural networks (Yang et al., 2024), stochastic formulations (Lin et al., 2025), and Sparse Gaussian Processes (Titsias, 2009; Burt et al., 2020). However, full Bayesian inference is computationally infeasible at LLM scale (Xue et al., 2021; Sankararaman et al., 2022), while efficient approximations like Laplace methods (Yang et al., 2024) apply calibration corrections *after* training. BLoB (Wang et al., 2024b) trains Bayesian LoRA weights via backpropagation but requires mean-field assumptions over the full LoRA parameter space.

We propose BAYESIAN-LORA, a calibration-aware fine-tuning framework derived from a structural observation: the Kronecker-factored conditional distribution in Sparse Gaussian Process (SGP) inference produces weight updates $M_W(U) = T_r\,U\,T_c$, which exhibit a *structural isomorphism* (in the functional sense of shared bilinear form, not strict algebraic equivalence) with LoRA's factorization $\Delta W = \frac{\alpha}{r}BA$. The projectors $T_r, T_c$ play analogous roles to LoRA's $B$, $A$ matrices, and the inducing variable $U$ replaces the deterministic core with a stochastic one. This suggests that *LoRA can be naturally embedded within a probabilistic framework*, where deterministic LoRA emerges as a limiting case. By maintaining a non-degenerate variational posterior over $U$, enriched by a normalizing flow (Lin et al.,

[1]School of Computer Science and Statistics, Trinity College Dublin, Dublin, Ireland and Lero the Research Ireland Centre for Software, Ireland [2]School of Computer Science, University College Dublin, Dublin, Ireland. Correspondence to: Goetz Botterweck <goetz.botterweck@tcd.ie>.

*Proceedings of the $43^{rd}$ International Conference on Machine Learning*, Seoul, South Korea. PMLR 306, 2026. Copyright 2026 by the author(s).

2026), we obtain a probabilistic generalization with calibrated uncertainty. This design offers three advantages: (i) uncertainty is modeled in a low-rank inducing space, keeping overhead minimal; (ii) a closed-form KL term avoids expensive Hessian computations; and (iii) calibration is optimized end-to-end during training.

We evaluate BAYESIAN-LORA on six commonsense reasoning benchmarks with Llama-2-7B, generative language modeling on WikiText-2, and mathematical reasoning with Qwen2.5-14B-Instruct and Qwen3-30B-A3B-Instruct-2507. Our work provides the following contributions:

- **Structural Isomorphism.** We identify a structural isomorphism (shared bilinear functional form, not strict algebraic equivalence) between Kronecker-factored SGP posteriors and LoRA's factorization. Under limiting conditions (point-mass posterior and vanishing conditional noise), BAYESIAN-LORA reduces to a deterministic bilinear update, motivating our probabilistic generalization.

- **Flow-Augmented Variational Inference.** We improve the expressiveness of the posterior by applying normalizing flows in the inducing space and derive an Evidence Lower Bound (ELBO) with a closed-form conditional KL term. This enables calibration-aware training with minimal overhead ($\approx 1.2\times$ training time, $\approx 0.42$M additional parameters).

- **Evaluation.** Tested across commonsense reasoning, text generation, and math tasks (including 14B dense and 30B Mixture-of-Experts (MoE) models), BAYESIAN-LORA improves NLL (Negative Log-Likelihood) and achieves strong calibration under distribution shift while maintaining competitive accuracy.

## 2. Related Work

Model calibration matters for trustworthy machine learning. Although modern neural networks can achieve high predictive accuracy, they are often poorly calibrated and produce overconfident predictions (Guo et al., 2017).

Post-hoc methods improve calibration by scaling model predictions after training (Kull et al., 2019; Guo et al., 2017). However, they cannot prevent weight-level uncertainty degradation, and under distribution shift, post-hoc calibration often becomes less effective (Desai & Durrett, 2020).

Bayesian methods quantify uncertainty by using BNNs (Izmailov et al., 2021; Lin et al., 2025; Kweon et al., 2025), Laplace approximations (Yang et al., 2024), and Gaussian Processes (Ranković & Schwaller, 2025), but they incur high costs at LLM scale. Full-model methods require approximate inference over entire parameter spaces (Graves,

2011; Blundell et al., 2015); Laplace methods depend on Hessians whose cost grows with model size (Daxberger et al., 2021; Ritter et al., 2018); GPs scale cubically in data (Seeger, 2004; Quiñonero-Candela & Rasmussen, 2005; Ritter et al., 2021); and ensembles multiply inference cost (Lakshminarayanan et al., 2017). Fully Factorized Gaussian over inducing weights (FFG-U) (Ritter et al., 2021) introduces inducing weights via a Kronecker-factorized decomposition of SGP weights, yielding a low-dimensional inducing weight matrix $U$. We are therefore inspired by FFG-U to model uncertainty within the low-rank subspace of LoRA. We further combine this with normalizing flows to enrich the posterior over $\Delta W$, improving its expressiveness with low overhead.

Parameter-Efficient Fine-Tuning (PEFT) introduces a small set of additional parameters while keeping the base LLM frozen, reducing fine-tuning cost and memory footprint. Methods include adapter-based tuning (Houlsby et al., 2019; Pfeiffer et al., 2021), prompt/prefix tuning (Lester et al., 2021; Li & Liang, 2021), and low-rank adaptation (LoRA and its variants) (Hu et al., 2022; Zhang et al., 2023; Dettmers et al., 2023). However, PEFT techniques employ deterministic updates and do not model uncertainty or optimize calibration objectives within the adapter subspace.

We integrate Sparse Gaussian Processes with LoRA by exploiting a structural isomorphism (shared bilinear form) between Kronecker-factored SGP posteriors and LoRA's factorization, bringing probabilistic inference to LLMs while maintaining PEFT-level efficiency.

## 3. Preliminaries

This section reviews LoRA (§3.1) and introduces the variational sparse inducing weight model (§3.2).

### 3.1. LoRA: Low-Rank Adaptation

Low-Rank Adaptation (LoRA) (Hu et al., 2022) parameterizes the weight update $\Delta W$ as a low-rank factorization $\frac{\alpha}{r}BA$ with rank $r \ll \min(d_{\text{in}}, d_{\text{out}})$, where $\alpha$ is a fixed scaling factor. The pre-trained weight $W_{\text{pre}} \in \mathbb{R}^{d_{\text{out}} \times d_{\text{in}}}$ remains frozen during training; only $B$ and $A$ are optimized:

$$\Delta W = \frac{\alpha}{r}BA, \qquad B \in \mathbb{R}^{d_{\text{out}} \times r}, \ A \in \mathbb{R}^{r \times d_{\text{in}}} \quad (1)$$

$$y = (W_{\text{pre}} + \Delta W)x = W_{\text{pre}}x + \frac{\alpha}{r}B(Ax) \quad (2)$$

where $x \in \mathbb{R}^{d_{\text{in}}}$ is the layer input and $y$ is the output. This reduces the number of trainable parameters from $d_{\text{in}}d_{\text{out}}$ (full fine-tuning) to $r(d_{\text{in}} + d_{\text{out}})$, and the adapter can be merged into $W_{\text{pre}}$ at inference time with no added latency. LoRA is typically applied to Transformer projection layers (e.g., query, key, value, and output projections; $W_q, W_k, W_v, W_o$).

However, the update $\Delta W$ is *deterministic*: it provides a single point estimate of the weight perturbation, capturing no information about uncertainty.

### 3.2. Variational Sparse Inducing Weight Model

**Core idea.** Rather than placing a distribution directly on $W \in \mathbb{R}^{d_{\text{out}} \times d_{\text{in}}}$, we introduce a compact *inducing matrix* $U \in \mathbb{R}^{r \times c}$ with $r \ll d_{\text{out}}$, $c \ll d_{\text{in}}$ that controls the distribution over $W$ (Ritter et al., 2021; Yang et al., 2024; Snelson & Ghahramani, 2005). As in Sparse GPs, the low-dimensional $U$ acts as a sufficient statistic for the posterior over $W$, keeping inference tractable.

**Prior and variational posterior on $U$.** We place a Gaussian (matrix-normal) prior on $U$:

$$p(U) = \mathcal{N}\big(\text{vec}(U) \mid \mathbf{0}, K_U\big), \quad K_U = K_c \otimes K_r \quad (3)$$

where $K_r \in \mathbb{R}^{r \times r}$ and $K_c \in \mathbb{R}^{c \times c}$ are learnable row and column covariance factors, and $\otimes$ denotes the Kronecker product. The variational posterior is:

$$q(U) = \mathcal{N}\big(\text{vec}(U) \mid \mathbf{m}, \mathbf{S}\big) \quad (4)$$

where $\mathbf{m}$ and $\mathbf{S}$ are the variational mean and covariance. Whitening can be applied so that the prior on $U$ becomes standard normal, simplifying the KL computation.

**Conditional distribution of $W$ given $U$.** Given the inducing variables $U$, the weight matrix $W$ follows a Gaussian conditional (Lin et al., 2025):

$$p(W \mid U) = \mathcal{N}\big(W \mid M_W(U), \Sigma_W\big) \quad (5)$$

where $\Sigma_W \in \mathbb{R}^{d_{\text{out}} d_{\text{in}} \times d_{\text{out}} d_{\text{in}}}$ is the prior conditional covariance. The covariance factors are parameterized as:

$$K_r = Z_r Z_r^\top + D_r^2, \qquad K_c = Z_c Z_c^\top + D_c^2 \quad (6)$$

with $Z_r \in \mathbb{R}^{r \times d_{\text{out}}}$, $Z_c \in \mathbb{R}^{c \times d_{\text{in}}}$ learnable, and $D_r \in \mathbb{R}^{r \times r}$, $D_c \in \mathbb{R}^{c \times c}$ diagonal noise matrices. The projection operators are

$$T_r = Z_r^\top K_r^{-1}, \qquad T_c = K_c^{-1} Z_c \quad (7)$$

where the subscripts $r$ and $c$ denote *row* and *column*, indicating that $T_r$ acts on the row space and $T_c$ on the column space of $U$. Together they define the conditional mean of $W$ as a bilinear projection (derivation in Appendix A):

$$M_W(U) = T_r U T_c \quad (8)$$

**Marginal distribution and ELBO.** The marginal distribution over $W$ is obtained by integrating out $U$:

$$q(W) = \int p(W \mid U) \, q(U) \, \mathrm{d}U \quad (9)$$

Optimizing only the low-dimensional variational parameters of $U$ thus suffices to approximate the high-dimensional posterior over $W$. The variational evidence lower bound takes the form:

$$\mathcal{L} = \mathbb{E}_{q(W)}\big[\log p(\mathcal{D} \mid W)\big] - \text{KL}\big(q(U) \,\|\, p(U)\big) - $$
$$\mathbb{E}_{q(U)}\Big[\text{KL}\big(q(W \mid U) \,\|\, p(W \mid U)\big)\Big] \quad (10)$$

where the three terms are: expected log-likelihood, KL over inducing variables, and conditional KL. The variational conditional has the same mean $M_W(U)$ as the prior conditional but a scaled covariance:

$$q(W \mid U) = \mathcal{N}\big(W \mid M_W(U), \lambda^2 \Sigma_W\big) \quad (11)$$

where $\lambda > 0$ is a learnable scale. Because $q(W \mid U)$ and $p(W \mid U)$ share the same mean and differ only by $\lambda$, the conditional KL admits a *closed-form* expression independent of $U$ (see §4).

**Proposition 3.1** (KL invariance under $T_\phi$). *Let $T_\phi$ be invertible and set $\Delta W = T_\phi(U)$. With pushforwards $q_\phi := T_{\phi\#} q_\psi$ and $p_\phi := T_{\phi\#} p$, we have*

$$\text{KL}\big(q_\phi \,\|\, p_\phi\big) = \text{KL}\big(q_\psi \,\|\, p\big) \quad (12)$$

*Hence, the ELBO written in $U$-space equals the ELBO written in $\Delta W$-space.* Proof. *See Appendix A.1.*

This result guarantees that we can optimize the ELBO in the compact $U$-space while the induced posterior over $\Delta W$ remains consistent.

## 4. Methodology: BAYESIAN-LORA

BAYESIAN-LORA replaces the deterministic low-rank update of LoRA with a probabilistic formulation. The LoRA weight matrices $B$ and $A$ are treated as random variables, each modeled by a per-layer inducing-variable posterior enriched by a normalizing flow; this architecture is illustrated in Figure 1. We describe how LoRA maps to this Bayesian formulation (§4.1), then derive the training objective (§4.2).

### 4.1. From LoRA to BAYESIAN-LORA

**Probabilistic low-rank update.** Recall that LoRA uses a deterministic rank-$r$ update:

$$\Delta W_{\text{LoRA}} = \tfrac{\alpha}{r} BA, \qquad B \in \mathbb{R}^{d_{\text{out}} \times r}, \; A \in \mathbb{R}^{r \times d_{\text{in}}} \quad (13)$$

In BAYESIAN-LORA, we replace this with a stochastic update. For each target layer, we introduce a low-dimensional inducing matrix $U \in \mathbb{R}^{r \times c}$ and "diffuse" its information into the high-dimensional weight space via the conditional Gaussian $p(W \mid U)$ (Eq. (5)). The conditional mean $M_W(U) = T_r U T_c$ (Eq. (8)) acts as a stochastic analog of the LoRA update $\frac{\alpha}{r} BA$: each Monte Carlo sample of $U$

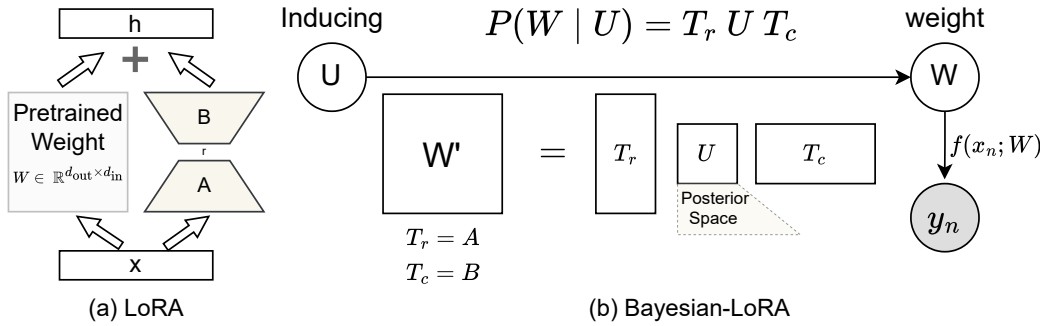

*Figure 1.* BAYESIAN-LORA overview. (a) Standard LoRA decomposes the weight update into low-rank matrices $B$ and $A$. (b) BAYESIAN-LORA places a Sparse Gaussian Process prior over the LoRA subspace. Inducing variables $U$ are transformed by a normalizing flow $T_\phi$ and projected via conditional Gaussians to produce stochastic LoRA matrices $B, A$. The effective weight $W_{\text{merged}} = W_{\text{pre}} + \frac{\alpha}{r}BA$ is used for the forward pass; Monte Carlo samples capture epistemic uncertainty.

produces a different weight realization, and the spread of these realizations encodes epistemic uncertainty.

Compared to LoRA's deterministic $\Delta W = \frac{\alpha}{r}BA$, BAYESIAN-LORA randomizes this update and quantifies uncertainty: $\Delta W$ is induced by the posterior uncertainty of $U$.

**Normalizing flow for posterior flexibility.** A purely Gaussian posterior on $U$ may be too restrictive for capturing weight-space uncertainty of LLMs. Following (Lin et al., 2026), we enrich the variational family while keeping the parameter count small by placing a normalizing flow (Rezende & Mohamed, 2015; Papamakarios et al., 2017) on top of a diagonal-Gaussian base distribution:

$$q_0(U_0) = \mathcal{N}\big(\text{vec}(U_0) \mid \mathbf{m}, \text{diag}(\boldsymbol{\sigma}^2)\big), \; \boldsymbol{\sigma} \in \mathbb{R}^{rc}_{>0},$$
$$U = T_\phi(U_0) \tag{14}$$

Here, $T_\phi$ is an invertible, differentiable map; in practice, we use a lightweight row-wise Masked Autoregressive Flow (MAF). When $T_\phi$ is the identity, $q_\phi(U)$ reduces to the diagonal-Gaussian baseline; increasing flow capacity improves posterior expressiveness and downstream calibration. The Gaussian prior on $U$ follows Eq. (3): $p(U) = \mathcal{N}\big(\text{vec}(U) \mid \mathbf{0}, K_c \otimes K_r\big)$.

### 4.2. Training Objective: Flow-Augmented ELBO

By the change-of-variables formula, the flow-transformed density of $U$ is:

$$\log q_\phi(U) = \log q_0\big(T_\phi^{-1}(U)\big) - \log\Big| \det J_{T_\phi}\big(T_\phi^{-1}(U)\big)\Big| \tag{15}$$

Substituting $q_\phi(U)$ into the standard SGP ELBO (Eq. (10)) yields:

$$\mathcal{L}_{\text{ELBO}} = \mathbb{E}_{U_0 \sim q_0, \, \varepsilon}\big[ \log p(\mathcal{D} \mid W)\big]$$
$$- \mathbb{E}_{U_0 \sim q_0}\Big[ \log q_0(U_0) - \log\big| \det J_{T_\phi}(U_0)\big|$$
$$- \log p\big(T_\phi(U_0)\big)\Big] - \frac{D}{2}\Big(\lambda^2 - 1 - 2\log \lambda\Big) \tag{16}$$

where $d_{W,\ell} = d_{\text{out},\ell}\, d_{\text{in},\ell}$ is the number of weight entries in the $\ell$-th replaced layer and $D = \sum_{\ell \in \mathcal{L}} d_{W,\ell}$ is the total across all such layers. The three terms correspond to: (1) the expected log-likelihood, approximated via Monte Carlo sampling; (2) the KL divergence over inducing variables $\text{KL}(q_\phi(U) \,\|\, p(U))$, which by Proposition 3.1 equals the KL in $\Delta W$-space; and (3) the conditional KL $\text{KL}(q(W \mid U) \,\|\, p(W \mid U))$, which has a closed form that is independent of $U$. See Appendix A.6 for detailed interpretations.

The Jacobian determinant $\log | \det J_{T_\phi}(U_0)|$ is computed efficiently by the autoregressive structure of the MAF.

**Structural isomorphism with LoRA.** The Kronecker decomposition $K_U = K_c \otimes K_r$ generates the conditional mean $M_W(U) = T_r U T_c$ (Eq. (8)), which exhibits a *structural isomorphism* with LoRA's $\Delta W = \frac{\alpha}{r}BA$ in the sense of shared bilinear functional form: the left projector $T_r \in \mathbb{R}^{d_{\text{out}} \times r}$ plays an analogous role to $B$, the right projector $T_c \in \mathbb{R}^{c \times d_{\text{in}}}$ to $A$, and the inducing variable $U \in \mathbb{R}^{r \times c}$ replaces the deterministic product with a stochastic low-rank core. When the inducing dimensions match the LoRA rank ($r = c = \text{LoRA rank}$), BAYESIAN-LORA produces weight updates in the same low-rank subspace while adding uncertainty quantification. We emphasize that this is a *functional* isomorphism (both produce low-rank bilinear weight updates), not a claim of strict algebraic equivalence. Standard LoRA directly optimizes $(B, A)$ as free parameters, whereas BAYESIAN-LORA optimizes $U$ with projection operators $T_r, T_c$ derived from the covariance structure.

The isomorphism holds at the level of the posterior mean. Concretely, the form of the weight update $T_r U T_c$ and the rank-$r$ low-rank structure are the same as LoRA's $BA$, so any function that LoRA can fit at rank $r$ can also be fitted by the posterior mean of BAYESIAN-LORA. What changes is how the parameters are learned. LoRA trains $B$ and $A$ directly with standard gradient descent. BAYESIAN-LORA

instead trains the inducing matrix $U$ by maximizing the ELBO, whose KL term pulls $U$ toward the GP prior $p(U)$. This is why the two methods reach similar accuracy but produce different confidence estimates: they fit functions from the same family, but follow different training paths.

**Corollary 4.1** (Deterministic limit). *Let $q(U) = \delta(U - U^*)$ be a point-mass posterior and $\lambda \to 0$. Then the* BAYESIAN-LORA *weight update reduces to the deterministic form $\Delta W = T_r U^* T_c$, recovering a bilinear low-rank update.*

*Proof.* Under $q(U) = \delta(U - U^*)$, sampling $U \sim q$ generates $U^*$ with probability one. The conditional mean (Eq. (8)) gives $M_W(U^*) = T_r U^* T_c$. As $\lambda \to 0$, the variational conditional noise $\lambda \Sigma_W^{1/2} \varepsilon \to 0$ (Eq. (11)), so $W = M_W(U^*)$ deterministically. $\square$

As illustrated in Figure 1, each Monte Carlo sample of $U$ produces a different weight realization through this bilinear structure, and the spread of realizations encodes epistemic uncertainty. The complete training procedure is given in Algorithm 1.

# 5. Experiments

We empirically evaluate BAYESIAN-LORA along three dimensions: (1) accuracy under a compute budget comparable to standard LoRA; (2) calibration and likelihood improvements; and (3) cost-effectiveness relative to existing Bayesian and post-hoc methods.

We use Llama-2-7B, Qwen2.5-14B-Instruct, and Qwen3-30B-A3B-Instruct-2507 as base backbones, applying LoRA adapters to the query (Q), key (K), and language model head weight matrices through the PEFT library (Mangrulkar et al., 2022). The corresponding linear layers are replaced with our Bayesian implementation (details in §4). We evaluate on six commonsense reasoning benchmarks (dataset details in Appendix D). All comparison methods share the same data splits, compute budget, and decoding settings. We apply label smoothing (Szegedy et al., 2016) with smoothing factor $\epsilon_{ls} = 0.1$ during training, which provides mild regularization and complements the Bayesian uncertainty modeling. Early stopping selects the checkpoint with the lowest validation NLL, following standard practice.[1] Each configuration is run with three random seeds; we report mean ± std for Accuracy (ACC ↑), Expected Calibration Error (ECE ↓; 15 bins), and Negative Log-Likelihood (NLL ↓) (see Appendix Table 10). For a fair comparison, all methods (including BAYESIAN-LORA) share the same fixed hyperparameters in Table 11; Bayesian optimization of BAYESIAN-LORA's learning rate and weight decay is a separate appendix analysis (Table 14).

---

[1]NLL-based early stopping is applied uniformly to all methods.

## 5.1. In-Distribution Evaluation

Table 1 compares BAYESIAN-LORA against eight baselines on six commonsense reasoning benchmarks. BAYESIAN-LORA achieves the highest accuracy on five of six benchmarks, with improvements of up to $+4.0$ points over standard LoRA. It also obtains the best ECE on two benchmarks (WinoGrande-S with an 84% reduction over Maximum A Posteriori (MAP), and WinoGrande-M) and the best NLL on four benchmarks. These improvements come from probabilistic training rather than post-hoc rescaling. Metric interpretations are in Appendix E.

**Generative language modeling.** Table 2 evaluates BAYESIAN-LORA on generative language modeling (WikiText-2). BAYESIAN-LORA (N = 2) matches or exceeds all baselines across NLL, Brier, and ECE on both validation and test splits. The improvement is pronounced on the top-5% most uncertain tokens, where BAYESIAN-LORA reduces test ECE from 1.60 (MAP) to 1.26. Unlike LA/LLLA, which require Hessian computation in generative settings, BAYESIAN-LORA estimates uncertainty end-to-end.

**Practical impact on downstream decisions.** Calibrated confidences directly affect any policy that uses model probabilities. In selective prediction, a better-calibrated model is more willing to skip examples that it is likely to get wrong, thereby improving the risk-coverage trade-off without changing the classifier. In confidence-based routing, low confidence can trigger a more expensive call, such as a larger model or verifier. This can help to reduce overconfidence, which would otherwise lead to wrong answers or silent failures. In safety-critical applications such as medical question answering, overconfident errors are far more costly than uncertain ones. Hence, the calibration gains in Tables 1 and 6 improve real decision quality, not just metrics.

## 5.2. Scaling to Larger Architectures

To demonstrate BAYESIAN-LORA's benefits beyond the 7B scale, we evaluate on mathematical reasoning (MATH) with **Qwen2.5-14B-Instruct** (a dense 14B model) and **Qwen3-30B-A3B-Instruct-2507** (hereafter **Qwen3-30B-A3B**; an MoE model with 30B total parameters but $\approx$ 3B active parameters per token).

Table 3 compares BAYESIAN-LORA against Baseline FT, Dropout, Temperature Scaling, LA (post-hoc), and BLoB. On Qwen2.5-14B-Instruct, BAYESIAN-LORA achieves CoT-NLL of 0.513 and CoT-ECE of 5.81, outperforming all baselines while improving accuracy to 51.1%. On Qwen3-30B-A3B, both NLL and ECE improve with no accuracy loss, confirming the $\mathcal{O}(rc)$ overhead remains negligible at scale.

---

**Algorithm 1** BAYESIAN-LORA (replace $B, A$): One Training Step

---

1: **Input:** Batch $\mathcal{B}$, target layers $\mathcal{L}$, rank $r$, scale $\alpha$, MC samples $N$, noise scale $\lambda$, flow $T_\phi$, base distribution $q_0$, per-layer inducing locations $(Z_r^{A/B}, Z_c^{A/B})$ and covariance factors $(K_r^{A/B}, K_c^{A/B})$
2: **Precompute:** For each layer $\ell \in \mathcal{L}$, compute:
3:     $T_r^A = (Z_r^A)^\top (K_r^A)^{-1}, \quad T_c^A = (K_c^A)^{-1} Z_c^A$        $\triangleright$ Projection operators for $A$, Eq. (7)
4:     $T_r^B = (Z_r^B)^\top (K_r^B)^{-1}, \quad T_c^B = (K_c^B)^{-1} Z_c^B$        $\triangleright$ Projection operators for $B$, Eq. (7)
5:     $\Sigma_{A,\ell}^{1/2}, \Sigma_{B,\ell}^{1/2}$        $\triangleright$ Covariance factors, Eq. (5)
6: **for** $\ell \in \mathcal{L}$ **do**
7:     **Sample inducing variables:** $U_0^{(1)}, \ldots, U_0^{(N)} \sim q_0(U_0)$        $\triangleright$ Base Gaussian, Eq. (14)
8:     **Apply flow transform:** $U^{(n)} \leftarrow T_\phi(U_0^{(n)})$ for $n = 1, \ldots, N$        $\triangleright$ Normalizing flow, Eq. (14)
9:     **for** $n = 1$ to $N$ **do**
10:        **Compute conditional means:**
11:        $\bar{A}_\ell^{(n)} = T_{r,\ell}^A U^{(n)} T_{c,\ell}^A$        $\triangleright$ Mean of $A$ given $U$, Eq. (8)
12:        $\bar{B}_\ell^{(n)} = T_{r,\ell}^B U^{(n)} T_{c,\ell}^B$        $\triangleright$ Mean of $B$ given $U$, Eq. (8)
13:        **Sample Gaussian noise:** $\varepsilon_A^{(n)}, \varepsilon_B^{(n)} \sim \mathcal{N}(0, I)$
14:        **Add conditional variance:**
15:        $A_\ell^{(n)} = \bar{A}_\ell^{(n)} + \lambda \Sigma_{A,\ell}^{1/2} \varepsilon_A^{(n)}$        $\triangleright$ Sample from $q(W_A \mid U)$, Eq. (11)
16:        $B_\ell^{(n)} = \bar{B}_\ell^{(n)} + \lambda \Sigma_{B,\ell}^{1/2} \varepsilon_B^{(n)}$        $\triangleright$ Sample from $q(W_B \mid U)$, Eq. (11)
17:        **Form LoRA update:** $\Delta W_\ell^{(n)} = \frac{\alpha}{r} B_\ell^{(n)} A_\ell^{(n)}$        $\triangleright$ Low-rank update, Eq. (1)
18:        **Effective weight:** $W_{\mathrm{merged},\ell}^{(n)} = W_{\mathrm{pre},\ell} + \Delta W_\ell^{(n)}$        $\triangleright$ Eq. (2)
19:     **end for**
20: **end for**
21: **Compute ELBO:**
22:     **Likelihood:** $\mathcal{L}_{\mathrm{data}} = \frac{1}{N} \sum_{n=1}^N \log p(\mathcal{B} \mid \{W_{\mathrm{merged},\ell}^{(n)}\}_{\ell \in \mathcal{L}})$        $\triangleright$ Expected log-likelihood, Eq. (16)
23:     **KL (inducing):** $\mathrm{KL}_U = \mathrm{KL}(q_\phi(U) \| p(U))$        $\triangleright$ KL in $U$-space, Eqs. (15)–(16)
24:     **KL (conditional):** $\mathrm{KL}_W = \frac{D}{2}(\lambda^2 - 1 - 2\log \lambda)$ where $D = \sum_{\ell \in \mathcal{L}} d_{\mathrm{out},\ell} \times d_{\mathrm{in},\ell}$        $\triangleright$ Closed-form, Eq. (31)
25:     $\mathcal{L}_{\mathrm{ELBO}} = \mathcal{L}_{\mathrm{data}} - \mathrm{KL}_U - \mathrm{KL}_W$        $\triangleright$ Eq. (16)
26: **Output:** $\mathcal{L}_{\mathrm{ELBO}}$ for gradient-based optimization

---

### 5.3. Efficiency Analysis

Table 4 compares computational overhead, normalized to standard LoRA (MAP):

- **Training memory.** Deep Ensembles require $\sim 3\times$ peak memory to train multiple independent models, which becomes prohibitive for 30B+ LLMs. BAYESIAN-LORA requires only $1.003\times$ peak memory to quantify uncertainty under the strict memory budget of a single model.

- **Training time.** BAYESIAN-LORA requires $1.229\times$ the training time of MAP, compared to BBB ($4.19\times$), Dropout ($4\times$), and Deep Ensembles ($3\times$).

- **Inference.** BAYESIAN-LORA supports two modes: (1) *deterministic mode* merges the posterior mean $W_{\mathrm{merged}} = W_{\mathrm{pre}} + \frac{\alpha}{r} T_r \mathbb{E}[U] T_c$ into the base weights with zero latency overhead, identical to standard LoRA; (2) *uncertainty mode* uses $N$ samples when calibrated confidence estimates are required. With N = 2, latency is $1.516\times$ MAP. All measurements were performed on the hardware described in Appendix F, Table 11.

- **Parameters.** BAYESIAN-LORA adds only 0.42M additional parameters (4.9M vs. 4.48M), the smallest overhead among all considered Bayesian methods.

### 5.4. Ablation Study

**Flow depth.** Table 5 ablates the normalizing flow depth $L$ on OBQA. Without any flow ($L=0$, pure SGP), accuracy drops by 2.6 points relative to $L=1$, confirming that the flow improves posterior expressiveness. A single flow layer ($L=1$) provides the best accuracy and a good accuracy–calibration trade-off. Deeper flows ($L = 2, 4$) further improve ECE at the cost of increased training time, while NLL is already near-optimal at $L = 1$. Hence, we adopt $L=1$ as the default.

Figure 2 varies the inducing-point dimension $r=c$ while keeping other hyperparameters fixed. The main experiments use $r=c=9$ to match the LoRA rank for a fair comparison. Increasing the inducing dimension improves calibration with diminishing returns beyond $r=16$; the parameter counts for each rank are compared in Appendix H, Table 16.

*Table 1.* Results on six commonsense reasoning benchmarks. We report Accuracy (ACC ↑), Expected Calibration Error (ECE ↓; 15 bins), and Negative Log-Likelihood (NLL ↓) at the early-stop checkpoint of each method. Values are mean ± std over three seeds. Inducing-point dimensions are set to $r = c = 9$ (matching the LoRA rank) for a fair parameter-count comparison.

| Metrics | Methods | WinoGrande-S | ARC-C | ARC-E | WinoGrande-M | OBQA | BoolQ |
|---|---|---|---|---|---|---|---|
| ACC ↑ | MAP (Hu et al., 2022) | $68.00 \pm 0.21$ | $64.90 \pm 1.10$ | $85.20 \pm 0.60$ | $73.70 \pm 0.90$ | $77.70 \pm 0.80$ | $85.80 \pm 0.40$ |
| | Dropout (Gal & Ghahramani, 2016) | $66.70 \pm 0.30$ | $64.90 \pm 1.90$ | $85.10 \pm 0.50$ | $73.50 \pm 0.90$ | $77.70 \pm 0.20$ | $85.90 \pm 0.40$ |
| | Ckpt Ens (Huang et al., 2017) | $66.70 \pm 0.30$ | $64.90 \pm 1.10$ | $85.20 \pm 0.60$ | $73.80 \pm 1.00$ | $78.20 \pm 0.20$ | $85.40 \pm 0.30$ |
| | Temp (Guo et al., 2017) | $67.00 \pm 0.60$ | $64.90 \pm 1.10$ | $85.20 \pm 0.60$ | $73.70 \pm 0.90$ | $77.70 \pm 0.80$ | $85.80 \pm 0.40$ |
| | BBB (Blundell et al., 2015) | $56.54 \pm 7.87$ | $68.13 \pm 1.27$ | $77.06 \pm 0.27$ | $73.63 \pm 2.44$ | $77.02 \pm 0.23$ | $83.17 \pm 0.53$ |
| | LLLA (post-hoc) (Yang et al., 2024) | $66.90 \pm 0.50$ | $66.10 \pm 0.60$ | $84.80 \pm 0.50$ | $73.70 \pm 0.90$ | $77.60 \pm 0.70$ | $85.80 \pm 0.40$ |
| | LA (post-hoc) (Yang et al., 2024) | $66.90 \pm 0.60$ | $66.90 \pm 1.10$ | $85.40 \pm 0.40$ | $73.70 \pm 1.00$ | $78.10 \pm 0.70$ | $85.80 \pm 0.40$ |
| | BLoB (N = 10) (Wang et al., 2024b) | $69.07 \pm 0.34$ | $68.81 \pm 1.09$ | $85.56 \pm 0.35$ | $73.69 \pm 0.17$ | $81.52 \pm 0.74$ | $\mathbf{86.99 \pm 0.24}$ |
| | BAYESIAN-LORA (N = 4) (End-to-End) | $\mathbf{70.90 \pm 0.10}$ | $\mathbf{68.90 \pm 0.20}$ | $\mathbf{85.90 \pm 0.30}$ | $\mathbf{74.30 \pm 0.20}$ | $\mathbf{81.60 \pm 0.10}$ | $86.10 \pm 0.20$ |
| ECE ↓ | MAP (Hu et al., 2022) | $30.80 \pm 1.80$ | $26.10 \pm 1.40$ | $8.90 \pm 0.30$ | $24.90 \pm 1.30$ | $9.80 \pm 1.00$ | $7.40 \pm 0.10$ |
| | Dropout (Gal & Ghahramani, 2016) | $29.50 \pm 1.60$ | $25.60 \pm 0.70$ | $8.80 \pm 0.60$ | $23.50 \pm 1.20$ | $8.80 \pm 0.80$ | $7.50 \pm 0.10$ |
| | Ckpt Ens (Huang et al., 2017) | $25.20 \pm 1.60$ | $26.10 \pm 1.40$ | $8.90 \pm 0.30$ | $22.80 \pm 1.40$ | $4.70 \pm 0.50$ | $3.20 \pm 0.50$ |
| | Temp (Guo et al., 2017) | $12.80 \pm 0.90$ | $\mathbf{4.60 \pm 1.00}$ | $4.70 \pm 0.80$ | $6.30 \pm 1.60$ | $7.20 \pm 2.60$ | $2.50 \pm 0.30$ |
| | BBB (Blundell et al., 2015) | $21.81 \pm 12.95$ | $26.23 \pm 1.47$ | $12.28 \pm 0.58$ | $15.76 \pm 4.71$ | $11.38 \pm 1.07$ | $3.74 \pm 0.10$ |
| | LLLA (post-hoc) (Yang et al., 2024) | $11.60 \pm 1.30$ | $5.60 \pm 2.10$ | $4.20 \pm 0.30$ | $3.80 \pm 1.40$ | $5.40 \pm 0.40$ | $1.70 \pm 0.50$ |
| | LA (post-hoc) (Yang et al., 2024) | $7.80 \pm 1.90$ | $7.50 \pm 1.20$ | $\mathbf{3.40 \pm 0.80}$ | $4.80 \pm 1.60$ | $\mathbf{3.50 \pm 0.40}$ | $1.90 \pm 0.30$ |
| | BLoB (N = 10) (Wang et al., 2024b) | $9.35 \pm 1.37$ | $9.59 \pm 1.88$ | $3.64 \pm 0.53$ | $3.01 \pm 0.12$ | $3.77 \pm 1.47$ | $\mathbf{1.41 \pm 0.19}$ |
| | BAYESIAN-LORA (N = 4) (End-to-End) | $\mathbf{4.90 \pm 0.20}$ | $9.20 \pm 0.70$ | $5.30 \pm 0.10$ | $\mathbf{3.00 \pm 0.10}$ | $5.70 \pm 0.20$ | $2.10 \pm 0.60$ |
| NLL ↓ | MAP (Hu et al., 2022) | $2.75 \pm 0.57$ | $1.64 \pm 0.19$ | $0.54 \pm 0.03$ | $2.43 \pm 0.50$ | $0.71 \pm 0.03$ | $0.43 \pm 0.01$ |
| | Dropout (Gal & Ghahramani, 2016) | $2.54 \pm 0.49$ | $1.55 \pm 0.16$ | $0.52 \pm 0.04$ | $2.12 \pm 0.35$ | $0.71 \pm 0.04$ | $0.43 \pm 0.01$ |
| | Ckpt Ens (Huang et al., 2017) | $1.31 \pm 0.04$ | $1.64 \pm 0.18$ | $0.54 \pm 0.03$ | $1.89 \pm 0.24$ | $0.65 \pm 0.02$ | $0.35 \pm 0.01$ |
| | Temp (Guo et al., 2017) | $0.68 \pm 0.01$ | $0.90 \pm 0.01$ | $0.43 \pm 0.02$ | $0.58 \pm 0.01$ | $0.67 \pm 0.02$ | $0.35 \pm 0.00$ |
| | BBB (Blundell et al., 2015) | $1.40 \pm 0.55$ | $2.23 \pm 0.04$ | $0.91 \pm 0.06$ | $0.84 \pm 0.15$ | $0.66 \pm 0.05$ | $0.31 \pm 0.00$ |
| | LLLA (post-hoc) (Yang et al., 2024) | $0.68 \pm 0.01$ | $0.94 \pm 0.02$ | $0.44 \pm 0.01$ | $0.56 \pm 0.01$ | $0.66 \pm 0.02$ | $0.35 \pm 0.00$ |
| | LA (post-hoc) (Yang et al., 2024) | $0.66 \pm 0.02$ | $0.86 \pm 0.02$ | $0.41 \pm 0.02$ | $0.55 \pm 0.01$ | $0.62 \pm 0.01$ | $0.34 \pm 0.00$ |
| | BLoB (N = 10) (Wang et al., 2024b) | $\mathbf{0.63 \pm 0.01}$ | $0.78 \pm 0.02$ | $0.40 \pm 0.01$ | $\mathbf{0.54 \pm 0.00}$ | $0.50 \pm 0.01$ | $0.31 \pm 0.00$ |
| | BAYESIAN-LORA (N = 4) (End-to-End) | $0.79 \pm 0.01$ | $\mathbf{0.77 \pm 0.05}$ | $\mathbf{0.38 \pm 0.09}$ | $0.62 \pm 0.11$ | $\mathbf{0.49 \pm 0.02}$ | $\mathbf{0.29 \pm 0.01}$ |

*Table 2.* Language modeling on **WikiText-2** (validation/test). We report NLL ↓, Brier score ↓, and ECE ↓ (15 bins) for both *all tokens* and the *top 5% most uncertain tokens* (selected by MAP predictive entropy).

| | All tokens | | | | | | Top 5% entropy tokens | | | | | |
|---|---|---|---|---|---|---|---|---|---|---|---|---|
| | Validation | | | Test | | | Validation | | | Test | | |
| Method | NLL ↓ | Brier ↓ | ECE ↓ | NLL ↓ | Brier ↓ | ECE ↓ | NLL ↓ | Brier ↓ | ECE ↓ | NLL ↓ | Brier ↓ | ECE ↓ |
| LoRA (MAP) | 1.76 | 0.52 | 1.68 | 1.75 | 0.52 | 1.48 | 5.16 | 0.97 | 0.82 | 5.16 | 0.97 | 1.60 |
| LoRA + Temp | 1.78 | 0.51 | 1.62 | 1.77 | 0.50 | 1.54 | 5.22 | 0.92 | 0.81 | 5.25 | 0.92 | 1.49 |
| Dropout (N = 4) | 1.77 | 0.50 | 1.54 | 1.70 | 0.50 | 1.59 | 5.21 | 0.94 | **0.79** | 5.19 | 0.95 | 1.51 |
| BAYESIAN-LORA (N = 1) | 1.73 | 0.51 | 1.46 | 1.71 | 0.51 | 1.51 | 5.14 | 0.95 | 0.80 | 5.14 | 0.92 | 1.31 |
| BAYESIAN-LORA (N = 2) | **1.70** | **0.48** | **1.36** | **1.66** | **0.48** | **1.41** | **5.13** | **0.91** | **0.79** | **5.12** | **0.90** | **1.26** |

**Empirical validation of MAP recovery (Corollary 4.1).**
Corollary 4.1 predicts that BAYESIAN-LORA reduces to standard LoRA when the posterior collapses to a point mass and the conditional noise vanishes. We verify this empirically on OBQA by training BAYESIAN-LORA with near-degenerate settings: $\lambda_{\text{init}} = 10^{-4}$, $\lambda_{\text{max}} = 10^{-4}$, $\sigma_{U,\text{max}} = 10^{-3}$, and flow depth $L = 0$ (identity transform). Table 7 compares this configuration against standard LoRA with MAP estimation.

The degenerate configuration matches standard LoRA, confirming that LoRA lies on the boundary of the BAYESIAN-

LORA posterior family. The full model improves all metrics by using calibrated uncertainty that the MAP limit discards.

### 5.5. Out-of-Distribution Robustness

Table 6 evaluates robustness under distribution shift (small: ARC-C/E; large: CS, Eng, Law, Health from MMLU). BAYESIAN-LORA achieves the best OoD accuracy on 5 of 6 shifted datasets.

For ECE, LA (post-hoc) leads *in-distribution* (ID) and *under small shifts* (ARC-C, ARC-E) by precise temperature tuning. *Under large shifts*, BAYESIAN-LORA achieves the best

*Table 3.* Performance on the **MATH** dataset with large-scale models. We report Chain-of-Thought NLL (CoT-NLL), CoT Expected Calibration Error (CoT-ECE), and final answer accuracy. Training hyperparameters are in Table 24.

| Model (Zero-shot) | Method | CoT-NLL ↓ | CoT-ECE ↓ | Answer Acc. ↑ |
|---|---|---|---|---|
| Qwen2.5-14B-Instruct | Baseline FT | 2.165 | 12.2 | 49.8 |
| | Dropout (Gal & Ghahramani, 2016) | 2.103 | 11.9 | 50.0 |
| | Temp (Guo et al., 2017) | 1.96 | 10.7 | 49.9 |
| | LA (post-hoc) (Yang et al., 2024) | 0.81 | 7.12 | 49.8 |
| | BLoB (N = 10) (Wang et al., 2024b) | 1.21 | 8.41 | 47.2 |
| | BAYESIAN-LORA (N = 4) | **0.513** | **5.81** | **51.1** |
| Qwen3-30B-A3B-Instruct-2507 | Baseline FT | 1.096 | 8.96 | 61.8 |
| | Temp (Guo et al., 2017) | 1.021 | 8.94 | 61.7 |
| | LA (post-hoc) (Yang et al., 2024) | 0.904 | 7.08 | 61.8 |
| | BLoB (N = 10) (Wang et al., 2024b) | 0.993 | 7.15 | 60.4 |
| | BAYESIAN-LORA (N = 4) | **0.721** | **6.32** | **61.9** |

*Table 4.* Efficiency comparison normalized to standard LoRA (MAP) on WinoGrande-M. All baseline results are reproduced from open-source code. Inference-cost comparisons use a comparable sample budget (N = 4) across sampling-based methods; the accuracy and calibration tables report BLoB at its recommended N = 10. KFAC denotes Kronecker-Factored Approximate Curvature.

| Method | Trainable Params | Train time (×MAP) | Peak mem. (×MAP) | Inference (×MAP) | Samples (Val) |
|---|---|---|---|---|---|
| MAP (LoRA) (Hu et al., 2022) | 4.48M | 1.00 | 1.00 | 1.00 | 1 |
| Dropout (Gal & Ghahramani, 2016) | 4.48M | ≈ 4× | ≈ 1× | ≈ 4× | 4 |
| Ckpt Ens (3) (Huang et al., 2017) | 1× MAP | ≈ 1× | ≈ 1× | ≈ 3× | 3 |
| Deep Ens (3) (Lakshminarayanan et al., 2017) | 3× MAP | ≈ 3× | ≈ 3× | ≈ 3× | 3 |
| BBB (Blundell et al., 2015) | ≈ 2× MAP | ≈ 4.19× | ≈ 1.05× | ≈ 4.6× | 4 |
| BLoB (N = 4) (Wang et al., 2024b) | ≈1.5×MAP | ≈ 1.11× | ≈ 0.95× | ≈ 6.3× | 4 |
| LLLA (post-hoc) (Yang et al., 2024) | 4.48M + KFAC (≈0.36M) | ≈ 1.052× | ≈ 1.002× | ≈ 1.9× | - |
| LA (post-hoc) (Yang et al., 2024) | 4.48M + KFAC (≈4.98M) | ≈ 1.117× | ≈ 1.004× | ≈ 4.36× | - |
| BAYESIAN-LORA | 4.9M | ≈ 1.229× | ≈ 1.003× | ≈ 1.516× / ≈ 2.790× | 2 / 4 |

*Table 5.* Ablation on flow depth $L$ in the posterior transform $T_\phi$ (OBQA). Values are macro-averages (mean $\pm$ std over three seeds). $\Delta$ denotes the change from the $L=1$ baseline. Efficiency is relative to standard LoRA (MAP).

| Flow depth $L$ | ACC ↑ | | ECE ↓ | | NLL ↓ | | Efficiency (× MAP) | |
|---|---|---|---|---|---|---|---|---|
| | value | $\Delta$ vs. $L=1$ | value | $\Delta$ vs. $L=1$ | value | $\Delta$ vs. $L=1$ | train time | peak mem. |
| 0 (pure SGP) | $79.0 \pm 0.21$ | -2.6 | $5.8 \pm 0.13$ | +0.1 | $0.58 \pm 0.08$ | +0.09 | 1.19 | 1.002 |
| 1 | $81.6 \pm 0.10$ | 0.0 | $5.7 \pm 0.20$ | 0.0 | $0.49 \pm 0.02$ | 0.00 | 1.23 | 1.003 |
| 2 | $80.8 \pm 0.14$ | -0.8 | $5.6 \pm 0.09$ | -0.1 | $0.52 \pm 0.06$ | +0.03 | 1.30 | 1.008 |
| 4 | $80.9 \pm 0.08$ | -0.7 | $4.9 \pm 0.03$ | -0.8 | $0.48 \pm 0.13$ | -0.01 | 1.38 | 1.010 |

ECE on 3 of 4 domains (CS, Law, Health); LA retains an advantage on Engineering. Post-hoc calibration learns a fixed rescaling that becomes stale under severe shift, while end-to-end training embeds uncertainty into the weights. BAYESIAN-LORA achieves the best NLL on 6 of 7 sets.

BAYESIAN-LORA and post-hoc methods are complementary: post-hoc excels when test data resembles the calibration set, while end-to-end Bayesian training is more robust under distributional mismatch.

See Appendix B for details on the analysis and Appendix H for hyperparameter optimization.

## 6. Conclusion

We identified a structural isomorphism (shared bilinear functional form) between Kronecker-factored SGP posteriors and LoRA's factorization, with BAYESIAN-LORA reducing to deterministic LoRA in the limit. Our method replaces LoRA's point estimate with a flow-augmented variational posterior for calibration-aware training. Across common-sense reasoning, language modeling, and math benchmarks at scales up to 14B (dense) and 30B (MoE), BAYESIAN-LORA achieves competitive accuracy with improved NLL. ECE gains are largest under distribution shift; post-hoc methods remain effective for small shifts.

*Table 6.* Robustness under distribution shift. We evaluate on the in-distribution OBQA test set and six out-of-distribution datasets spanning small shifts (ARC-C, ARC-E) and large shifts (CS, Eng, Law, Health from MMLU). Experimental settings follow (Yang et al., 2024).

| | | ID | Smaller Distribution Shift | | Larger Distribution Shift | | | |
|---|---|---|---|---|---|---|---|---|
| Metrics | Methods | OBQA | ARC-C | ARC-E | CS | Eng | Law | Health |
| ACC ↑ | MAP (Hu et al., 2022) | $77.7 \pm 0.8$ | $67.9 \pm 1.4$ | $77.7 \pm 0.3$ | $42.0 \pm 3.2$ | $41.2 \pm 2.0$ | $37.4 \pm 0.4$ | $48.3 \pm 0.3$ |
| | Dropout (Gal & Ghahramani, 2016) | $77.7 \pm 0.2$ | $67.7 \pm 0.6$ | $77.2 \pm 0.6$ | $41.9 \pm 2.2$ | $39.6 \pm 1.7$ | $37.9 \pm 0.4$ | $48.2 \pm 0.9$ |
| | Ckpt Ens (Huang et al., 2017) | $78.2 \pm 0.2$ | $67.9 \pm 0.8$ | $77.4 \pm 0.8$ | $41.1 \pm 2.0$ | $38.7 \pm 1.2$ | $37.7 \pm 0.2$ | $48.2 \pm 0.6$ |
| | Temp (Guo et al., 2017) | $77.7 \pm 0.8$ | $68.0 \pm 0.2$ | $76.7 \pm 1.0$ | $43.5 \pm 0.9$ | $\mathbf{44.4 \pm 2.0}$ | $37.4 \pm 0.1$ | $47.7 \pm 0.8$ |
| | BBB (Blundell et al., 2015) | $77.0 \pm 0.2$ | $67.3 \pm 1.2$ | $75.8 \pm 0.8$ | $40.5 \pm 0.3$ | $36.7 \pm 0.2$ | $36.1 \pm 0.2$ | $47.6 \pm 0.5$ |
| | BLoB(N = 10) (Wang et al., 2024b) | $81.5 \pm 0.7$ | $67.7 \pm 1.1$ | $76.3 \pm 0.8$ | $44.6 \pm 0.4$ | $42.3 \pm 1.7$ | $37.4 \pm 1.2$ | $47.3 \pm 1.5$ |
| | LLLA (post-hoc) (Yang et al., 2024) | $77.6 \pm 0.7$ | $68.1 \pm 0.0$ | $78.1 \pm 0.0$ | $45.6 \pm 0.0$ | $38.9 \pm 0.0$ | $37.1 \pm 0.0$ | $48.5 \pm 0.0$ |
| | LA (post-hoc) (Yang et al., 2024) | $78.1 \pm 0.7$ | $69.2 \pm 0.0$ | $78.5 \pm 0.0$ | $45.1 \pm 0.0$ | $39.1 \pm 0.0$ | $37.3 \pm 0.0$ | $49.1 \pm 0.0$ |
| | BAYESIAN-LORA (N = 4) (End-to-End) | $\mathbf{81.6 \pm 0.1}$ | $\mathbf{69.5 \pm 0.1}$ | $\mathbf{78.9 \pm 0.2}$ | $\mathbf{46.3 \pm 0.1}$ | $37.5 \pm 0.2$ | $\mathbf{37.9 \pm 0.1}$ | $\mathbf{50.6 \pm 0.1}$ |
| ECE ↓ | MAP (Hu et al., 2022) | $9.80 \pm 1.0$ | $22.2 \pm 1.2$ | $15.8 \pm 1.0$ | $34.2 \pm 3.1$ | $38.4 \pm 1.7$ | $35.2 \pm 0.7$ | $34.2 \pm 0.8$ |
| | Dropout (Gal & Ghahramani, 2016) | $8.80 \pm 0.8$ | $21.4 \pm 0.4$ | $15.5 \pm 1.0$ | $33.7 \pm 2.0$ | $38.6 \pm 2.9$ | $34.2 \pm 0.6$ | $33.5 \pm 0.2$ |
| | Ckpt Ens (Huang et al., 2017) | $4.70 \pm 0.5$ | $17.7 \pm 0.7$ | $12.1 \pm 0.6$ | $29.1 \pm 2.3$ | $32.5 \pm 1.8$ | $32.1 \pm 0.1$ | $29.0 \pm 0.3$ |
| | Temp (Guo et al., 2017) | $7.20 \pm 2.6$ | $9.41 \pm 3.5$ | $6.33 \pm 1.3$ | $16.4 \pm 5.5$ | $16.1 \pm 4.6$ | $22.7 \pm 5.8$ | $17.4 \pm 6.0$ |
| | BBB (Blundell et al., 2015) | $11.4 \pm 1.1$ | $19.9 \pm 0.7$ | $13.4 \pm 0.9$ | $18.6 \pm 1.6$ | $22.4 \pm 3.1$ | $28.1 \pm 2.5$ | $27.4 \pm 1.8$ |
| | BLoB (N = 10) (Wang et al., 2024b) | $3.77 \pm 1.5$ | $9.55 \pm 0.4$ | $5.48 \pm 1.3$ | $12.6 \pm 1.7$ | $22.3 \pm 1.9$ | $25.3 \pm 2.1$ | $16.4 \pm 1.6$ |
| | LLLA (post-hoc) (Yang et al., 2024) | $5.40 \pm 0.4$ | $21.3 \pm 0.0$ | $14.8 \pm 0.0$ | $30.3 \pm 0.0$ | $39.7 \pm 0.0$ | $33.6 \pm 0.0$ | $33.5 \pm 0.0$ |
| | LA (post-hoc) (Yang et al., 2024) | $\mathbf{3.50 \pm 0.4}$ | $\mathbf{5.50 \pm 0.0}$ | $\mathbf{3.10 \pm 0.0}$ | $14.5 \pm 0.0$ | $\mathbf{12.8 \pm 0.0}$ | $23.9 \pm 0.0$ | $17.6 \pm 0.0$ |
| | BAYESIAN-LORA (N = 4) (End-to-End) | $5.70 \pm 0.2$ | $8.10 \pm 0.1$ | $5.21 \pm 0.1$ | $\mathbf{11.1 \pm 0.0}$ | $20.4 \pm 0.1$ | $\mathbf{16.5 \pm 0.0}$ | $\mathbf{12.9 \pm 0.0}$ |
| NLL ↓ | MAP (Hu et al., 2022) | $0.71 \pm 0.03$ | $1.30 \pm 0.07$ | $1.04 \pm 0.10$ | $1.90 \pm 0.12$ | $2.19 \pm 0.15$ | $2.12 \pm 0.03$ | $2.09 \pm 0.08$ |
| | Dropout (Gal & Ghahramani, 2016) | $0.71 \pm 0.04$ | $1.24 \pm 0.06$ | $1.01 \pm 0.09$ | $1.86 \pm 0.10$ | $2.14 \pm 0.13$ | $2.09 \pm 0.02$ | $2.05 \pm 0.07$ |
| | Ckpt Ens (Huang et al., 2017) | $0.65 \pm 0.02$ | $1.03 \pm 0.03$ | $0.80 \pm 0.03$ | $1.55 \pm 0.04$ | $1.72 \pm 0.01$ | $1.94 \pm 0.01$ | $1.74 \pm 0.02$ |
| | Temp (Guo et al., 2017) | $0.67 \pm 0.02$ | $0.90 \pm 0.05$ | $0.66 \pm 0.01$ | $1.31 \pm 0.06$ | $1.32 \pm 0.07$ | $1.65 \pm 0.16$ | $1.36 \pm 0.10$ |
| | BBB (Blundell et al., 2015) | $0.66 \pm 0.05$ | $1.06 \pm 0.01$ | $0.79 \pm 0.02$ | $1.49 \pm 0.05$ | $1.83 \pm 0.08$ | $1.65 \pm 0.20$ | $1.29 \pm 0.12$ |
| | BLoB (N = 10) (Wang et al., 2024b) | $0.50 \pm 0.01$ | $0.83 \pm 0.01$ | $\mathbf{0.60 \pm 0.01}$ | $1.38 \pm 0.01$ | $1.81 \pm 0.10$ | $2.01 \pm 0.09$ | $1.94 \pm 0.08$ |
| | LLLA (post-hoc) (Yang et al., 2024) | $0.66 \pm 0.02$ | $0.88 \pm 0.00$ | $0.64 \pm 0.00$ | $1.28 \pm 0.00$ | $1.27 \pm 0.00$ | $1.49 \pm 0.00$ | $1.31 \pm 0.00$ |
| | LA (post-hoc) (Yang et al., 2024) | $0.62 \pm 0.01$ | $0.85 \pm 0.00$ | $0.62 \pm 0.00$ | $1.26 \pm 0.00$ | $1.27 \pm 0.00$ | $1.68 \pm 0.00$ | $1.35 \pm 0.00$ |
| | BAYESIAN-LORA (N = 4) (End-to-End) | $\mathbf{0.49 \pm 0.02}$ | $\mathbf{0.82 \pm 0.04}$ | $0.80 \pm 0.04$ | $\mathbf{1.22 \pm 0.04}$ | $\mathbf{1.26 \pm 0.02}$ | $\mathbf{1.42 \pm 0.04}$ | $\mathbf{1.20 \pm 0.02}$ |

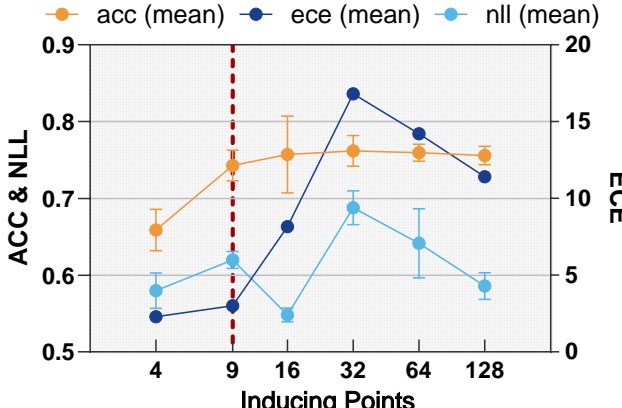

*Figure 2.* Ablation on inducing-point dimension $r = c$. Increasing the dimension improves calibration (lower ECE) with diminishing returns beyond $r=16$. The dashed line shows the default $r = c = 9$ used in the main-paper BAYESIAN-LORA experiments.

*Table 7.* MAP recovery validation on OBQA. Degenerate BAYESIAN-LORA uses near-zero posterior variance and no flow, approximating the MAP limit of Corollary 4.1. Results are mean ± std over three seeds.

| Method | ACC ↑ | ECE ↓ | NLL ↓ |
|---|---|---|---|
| LoRA (MAP) | $77.70 \pm 0.80$ | $9.80 \pm 1.00$ | $0.71 \pm 0.03$ |
| Degenerate BAYESIAN-LORA | $77.81 \pm 0.71$ | $9.77 \pm 1.02$ | $0.70 \pm 0.02$ |
| BAYESIAN-LORA (full) | $81.60 \pm 0.10$ | $5.70 \pm 0.20$ | $0.49 \pm 0.02$ |

**Limitations.** Our per-layer inducing matrices are modeled independently; hierarchical priors could capture inter-layer correlations. The Bayesian optimization analysis (Table 14) shows accuracy-calibration trade-offs with objective-dependent optimal hyperparameters. Extensions to other modalities and instruction-tuning/RLHF remain future work, as do tighter theoretical bounds on the sparse inducing approximation.

## Acknowledgements

This publication is based on research funded by the European Union's Horizon Europe 2021–2027 framework programme, Marie Skłodowska-Curie Actions, Grant Agreement No. 101072456; Taighde Éireann – Research Ireland under grant number 13/RC/2094_2 to Lero the Research Ireland Centre for Software and grant number 12/RC/2289_P2 to Insight Centre for Data Analytics; and the European Research Council (ERC) under the Horizon 2020 research and innovation programme (Grant Agreement No. 884951).

## Impact Statement

This work improves calibration and uncertainty quantification for fine-tuned LLMs. Well-calibrated models are essential for safe deployment in high-stakes domains such as medical diagnosis and autonomous systems, helping users identify when model outputs may be unreliable. We do not foresee negative societal consequences unique to this work.

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

# A. Derivations for the Variational Sparse Inducing Weight Model

## A.1. Model Specification and Joint Density

We define $W \in \mathbb{R}^{d_{\text{out}} \times d_{\text{in}}}$ as the layer weight matrix and $U \in \mathbb{R}^{r \times c}$ the low-dimensional inducing matrix with $r \ll d_{\text{out}}$ and $c \ll d_{\text{in}}$. We place a Gaussian (equivalently, matrix-normal) prior on $U$:

$$p(U) = \mathcal{N}\big(\text{vec}(U) \mid \mathbf{0}, K_U\big), \ K_U = K_c \otimes K_r \quad (17)$$

which is equivalent to $U \sim \mathcal{MN}(\mathbf{0}, K_r, K_c)$. Given $U$, the prior conditional distribution of $W$ is Gaussian

$$p(W \mid U) = \mathcal{N}\big(W \mid M_W(U), \Sigma_W\big) \quad (18)$$

where $M_W(U)$ is linear in $U$. The variational conditional shares the same mean but has a scaled covariance,

$$q(W \mid U) = \mathcal{N}\big(W \mid M_W(U), \lambda^2 \Sigma_W\big), \quad (19)$$

where $\lambda > 0$ is a learnable scale parameter. The row and column covariance factors are parameterized as

$$K_r = Z_r Z_r^\top + D_r^2, \qquad K_c = Z_c Z_c^\top + D_c^2 \quad (20)$$

The likelihood factorizes over data $\mathcal{D} = \{(x_n, y_n)\}_{n=1}^N$ as $p(\mathcal{D} \mid W) = \prod_{n=1}^N p(y_n \mid f(x_n; W))$. The joint density is:

$$p(U, W, \mathcal{D}) = p(U) \, p(W \mid U) \, p(\mathcal{D} \mid W) \quad (21)$$

## A.2. Variational Family and Factorization

We posit a Gaussian variational posterior on $U$,

$$q(U) = \mathcal{N}\big(\text{vec}(U) \mid \mathbf{m}, \mathbf{S}\big) \quad (22)$$

As a starting point, we first adopt the classical SGP choice of keeping the model conditional $p(W \mid U)$ inside the variational family:

$$q(U, W) = q(U) \, p(W \mid U)$$

We later relax this to the scaled variational conditional $q(W \mid U) = \mathcal{N}(M_W(U), \lambda^2 \Sigma_W)$ of Eq. 19, which reinstates the conditional KL term (see Eq.26 and the discussion thereafter). Marginalizing $U$ gives the variational distribution over $W$:

$$q(W) = \int p(W \mid U) \, q(U) \, \mathrm{d}U \quad (23)$$

## A.3. ELBO Derivation

Starting from $\log p(\mathcal{D}) = \log \int p(U, W, \mathcal{D}) \, \mathrm{d}U \, \mathrm{d}W$ and inserting $q(U, W)$ (Ritter et al., 2021):

$$\log p(\mathcal{D}) = \\ \log \int q(U, W) \frac{p(U) \, p(W \mid U) \, p(\mathcal{D} \mid W)}{q(U) \, p(W \mid U)} \, \mathrm{d}U \, \mathrm{d}W \quad (24)$$

Jensen's inequality yields the ELBO

$$\mathcal{L} = \mathbb{E}_{q(U)p(W|U)}\Big[ \log p(\mathcal{D} \mid W) \Big] - \\ \mathbb{E}_{q(U)}\Big[ \log \tfrac{q(U)}{p(U)} \Big] \quad (25)$$

i.e.

$$\mathcal{L} = \mathbb{E}_{q(W)}\Big[ \log p(\mathcal{D} \mid W) \Big] - \text{KL}\big(q(U) \,\|\, p(U)\big) \quad (26)$$

The form in Eq. (26) corresponds to the classical SGP setting in which $q(W \mid U) = p(W \mid U)$, so the conditional contribution vanishes. In our case the variational and prior conditionals share the same mean but differ by the scale $\lambda$, i.e. $q(W \mid U) = \mathcal{N}(M_W(U), \lambda^2 \Sigma_W)$ versus $p(W \mid U) = \mathcal{N}(M_W(U), \Sigma_W)$, so this term no longer cancels and must be retained. To make it explicit we use the KL chain rule for the joint distribution $q(W, U) = q(U) \, q(W \mid U)$:

$$\text{KL}\big(q(W, U) \,\|\, p(W, U)\big) = \text{KL}\big(q(U) \,\|\, p(U)\big) \\ + \mathbb{E}_{q(U)}\Big[\text{KL}\big(q(W \mid U) \,\|\, p(W \mid U)\big)\Big] \quad (27)$$

Substituting this decomposition into Eq. (26), i.e. adding and subtracting $\mathbb{E}_{q(U)}[\log p(W \mid U)]$ inside the expectation, separates the joint KL into a marginal term over $U$ and a conditional term over $W \mid U$:

$$\mathcal{L} = \mathbb{E}_{q(W)}\Big[ \log p(\mathcal{D} \mid W) \Big] - \text{KL}\big(q(U) \,\|\, p(U)\big) \\ - \mathbb{E}_{q(U)}\Big[\text{KL}\big(q(W \mid U) \,\|\, p(W \mid U)\big)\Big] \quad (28)$$

which matches the presentation in the main text. The conditional KL in the last line is closed-form and independent of $U$, as shown in Eq. (31).

## A.4. Closed-Form Terms

**KL between Gaussians for $q(U)$ and $p(U)$.** Define $d_U = rc$ and denote $\mathbf{m} \in \mathbb{R}^{d_U}, \mathbf{S} \in \mathbb{R}^{d_U \times d_U}$, and $K_U = K_c \otimes K_r$. Then

$$\text{KL}\big(q(U) \,\|\, p(U)\big) = \frac{1}{2}\Big(\text{tr}(K_U^{-1}\mathbf{S}) + \\ \mathbf{m}^\top K_U^{-1}\mathbf{m} - d_U + \log \tfrac{|K_U|}{|\mathbf{S}|}\Big) \quad (29)$$

Using $\log |\mathbf{K}_c \otimes \mathbf{K}_r| = c \log |\mathbf{K}_r| + r \log |\mathbf{K}_c|$ simplifies evaluation. In the whitened case with $p(U) = \mathcal{N}(\mathbf{0}, \mathbf{I}_{d_U})$,

$$\text{KL}\big(q(U) \,\|\, p(U)\big) = \\ \frac{1}{2}\Big(\text{tr}(\mathbf{S}) + \mathbf{m}^\top \mathbf{m} - d_U - \log |\mathbf{S}|\Big) \quad (30)$$

**Conditional KL for $q(W \mid U)$ vs. $p(W \mid U)$.** Both share the same mean $M_W(U)$. The variational conditional has covariance $\Sigma_q = \lambda^2 \Sigma_W$ (from Eq. (19)), and the prior

conditional has covariance $\Sigma_p = \Sigma_W$ (from Eq. (18)). Set $d_W = d_{\text{out}}d_{\text{in}}$. Then

$$
\begin{aligned}
\text{KL}&\big(q(W \mid U) \,\|\, p(W \mid U)\big) \\
&= \frac{1}{2}\Big(\text{tr}(\Sigma_p^{-1}\Sigma_q) - d_W + \log\frac{|\Sigma_p|}{|\Sigma_q|}\Big) \\
&= \frac{1}{2}\Big(\lambda^2 d_W - d_W - 2d_W \log\lambda\Big) \\
&= \frac{d_W}{2}\big(\lambda^2 - 1 - 2\log\lambda\big)
\end{aligned}
\tag{31}
$$

This is independent of $U$ and thus the expectation $\mathbb{E}_{q(U)}[\cdot]$ in (28) is trivial.

### A.5. Form of $q(W)$: Mean and Covariance

Vectorizing $W$ and using $\text{vec}(T_r U T_c) = (T_c^\top \otimes T_r)\,\text{vec}(U)$, one obtains

$$
\begin{aligned}
\mathbb{E}_q[\,\text{vec}(W)\,] &= (T_c^\top \otimes T_r)\,\mathbf{m}, \\
\mathbb{E}_q[\,W\,] &= T_r\,M\,T_c
\end{aligned}
\tag{32}
$$

where $M$ is the matricized version of $\mathbf{m}$. Moreover, since $\Sigma_q(W \mid U) = \lambda^2 \Sigma_W$ is independent of $U$,

$$
\text{Cov}_q[\,\text{vec}(W)\,] = \lambda^2 \Sigma_W + (T_c^\top \otimes T_r)\,\mathbf{S}\,(T_c \otimes T_r^\top) \tag{33}
$$

Thus $q(W)$ is Gaussian with the above mean and covariance whenever $q(U)$ is Gaussian and $M_W(U)$ is linear in $U$.

Finally,

$$
\log p(\mathcal{D}) = \mathcal{L} + \text{KL}\big(q(U) \,\|\, p(U \mid \mathcal{D})\big), \tag{34}
$$

So maximizing $\mathcal{L}$ minimizes the divergence from the variational posterior to the exact posterior over $U$.

**Proposition A.1** (Details of $U$-space-independent KL). *Let $U \in \mathbb{R}^{d_U}$ denote the inducing variables (with $d_U = rc$) with a Gaussian prior $p(U) = \mathcal{N}(\mu_p, \Sigma_p)$ and variational posterior $q_\psi(U) = \mathcal{N}(\mu_q, \Sigma_q)$. Let $T_\phi : \mathbb{R}^{d_U} \to \mathbb{R}^{d_U}$ be an invertible $C^1$ map (the adapter flow) with Jacobian $J_{T_\phi}(U)$, and define the LoRA parameters as the deterministic pushforward $\Delta W = T_\phi(U)$. For data $\mathcal{D}$ with likelihood $p(\mathcal{D} \mid \Delta W)$, the ELBO in $U$-space*

$$
\begin{aligned}
\mathcal{L}(\phi, \psi) = \mathbb{E}_{q_\psi(U)}\big[\log p(\mathcal{D} \mid T_\phi(U))\big] \\
- \text{KL}\big(q_\psi(U) \,\|\, p(U)\big)
\end{aligned}
\tag{35}
$$

*is equivalent to the ELBO in $\Delta W$-space,*

$$
\begin{aligned}
\mathcal{L}(\phi, \psi) = \mathbb{E}_{q_\phi(\Delta W)}\big[\log p(\mathcal{D} \mid \Delta W)\big] \\
- \text{KL}\big(q_\phi(\Delta W) \,\|\, p_\phi(\Delta W)\big)
\end{aligned}
\tag{36}
$$

*where $q_\phi(\Delta W) := T_{\phi\#}q_\psi$ and $p_\phi(\Delta W) := T_{\phi\#}p$ are pushforwards under $T_\phi$. Moreover, the KL term is invariant under $T_\phi$:*

$$
\text{KL}\big(q_\phi(\Delta W) \,\|\, p_\phi(\Delta W)\big) = \text{KL}\big(q_\psi(U) \,\|\, p(U)\big) \tag{37}
$$

*In particular, when $q_\psi$ and $p$ are Gaussian, the KL admits a closed form:*

$$
\begin{aligned}
\text{KL}\big(\mathcal{N}(\mu_q, \Sigma_q) \,\|\, \mathcal{N}(\mu_p, \Sigma_p)\big) = \frac{1}{2}\bigg( \text{tr}(\Sigma_p^{-1}\Sigma_q) \\
+ (\mu_p - \mu_q)^\top \Sigma_p^{-1}(\mu_p - \mu_q) \\
- d_U + \log\frac{\det\Sigma_p}{\det\Sigma_q}\bigg)
\end{aligned}
\tag{38}
$$

*Proof.* Since $\Delta W = T_\phi(U)$ is a deterministic, invertible change of variables, the data term satisfies $\mathbb{E}_{q_\psi(U)}[\log p(\mathcal{D} \mid T_\phi(U))] = \mathbb{E}_{q_\phi(\Delta W)}[\log p(\mathcal{D} \mid \Delta W)]$. For the KL term, write the pushforward densities via change of variables: $q_\phi(\Delta W) = q_\psi(U)\,|\det J_{T_\phi}(U)|^{-1}$ and $p_\phi(\Delta W) = p(U)\,|\det J_{T_\phi}(U)|^{-1}$ with $\Delta W = T_\phi(U)$. Then

$$
\begin{aligned}
\text{KL}(q_\phi \| p_\phi) &= \int q_\phi(\Delta W) \log\frac{q_\phi(\Delta W)}{p_\phi(\Delta W)}\,\mathrm{d}\Delta W \\
&= \int q_\psi(U) \log\frac{q_\psi(U)}{p(U)}\,\mathrm{d}U = \text{KL}(q_\psi \| p)
\end{aligned}
\tag{39}
$$

where the Jacobian determinants cancel exactly. Combining the two parts establishes the equivalent ELBO forms (35)–(36) and the invariance (37); the KL does not depend on $\Delta W$ nor on $T_\phi$. When $q_\psi$ and $p$ are Gaussian, (38) follows from the standard closed-form KL between multivariate Gaussians. $\square$

### A.6. Interpretation of ELBO Terms

The ELBO in Eq. (16) consists of three terms with the following interpretations:

1. **Expected log-likelihood.** The first term $\mathbb{E}_{U_0 \sim q_0,\,\varepsilon}[\log p(\mathcal{D} \mid W)]$ represents the expected data fit. This is approximated by Monte Carlo sampling: we draw $N$ inducing samples from $q_0$, map each through the conditional mean $M_W(U)$ and add scaled Gaussian noise (controlled by $\lambda$) to obtain weight realizations, then average the log-likelihoods over these samples.

2. **KL over inducing variables.** The second term

$$
\begin{aligned}
\mathbb{E}_{U_0 \sim q_0}\Big[\log q_0(U_0) - \log\big|\det J_{T_\phi}(U_0)\big| \\
- \log p\big(T_\phi(U_0)\big)\Big]
\end{aligned}
\tag{40}
$$

computes $\text{KL}(q_\phi(U) \,\|\, p(U))$ via the flow density (using the change-of-variables formula). By Proposition 3.1, this KL is invariant under the transformation $T_\phi$, meaning it equals the KL in $\Delta W$-space. Therefore, we may optimize in $U$-space while inducing a consistent posterior over weights.

3. **Conditional KL (closed-form).** The third term $\frac{D}{2}(\lambda^2 - 1 - 2\log\lambda)$ represents $\mathrm{KL}(q(W \mid U) \| p(W \mid U))$. Since $q(W \mid U)$ and $p(W \mid U)$ share the same conditional mean and differ only in their covariances ($\lambda^2 \Sigma_W$ vs. $\Sigma_W$), the conditional KL reduces to this closed form, which is *independent of $U$* and requires no sampling or Hessian computation. The derivation is provided in Appendix A.5.

## B. Analysis of Out-of-Distribution Robustness

Table 6 evaluates robustness under distribution shift, covering small shifts (ARC-C, ARC-E) and large shifts (CS, Eng, Law, Health from MMLU). We highlight three key findings:

**Accuracy under shift.** BAYESIAN-LoRA achieves the best OoD accuracy on 5 of 6 shifted datasets (ARC-C, ARC-E, CS, Law, Health) and the best ID accuracy (OBQA). The largest OoD accuracy gain is $+1.5$ points on Health over the next-best method (LA). This suggests that the end-to-end Bayesian training acts as a form of distributional regularization that prevents overfitting to the in-distribution data.

**Calibration under shift.** While LA achieves the best ECE in-distribution (ID) and under small shifts (ARC-C/E), BAYESIAN-LoRA achieves the best ECE on three of four large-shift domains: CS (11.1), Law (16.5), and Health (12.9). However, LA retains an advantage on Engineering (12.8 vs. 20.4), indicating that the relative benefit of end-to-end Bayesian training varies across OoD domains. This is a critical distinction: post-hoc methods calibrate well on data similar to the training set but degrade under severe distribution shift, whereas end-to-end Bayesian training produces more robust uncertainty estimates.

**Likelihood under shift.** BAYESIAN-LoRA achieves the best NLL on 6 of 7 evaluation sets (all except ARC-E OoD). The NLL improvements under large shifts are substantial: e.g., 1.20 vs. 1.29 (BBB) on Health, 1.42 vs. 1.49 (LLLA) on Law. Combined with the efficiency results (Table 4), BAYESIAN-LoRA achieves OoD performance comparable to or better than computationally heavy ensembles at a fraction of the cost.

## C. Effect of Monte Carlo Samples on Accuracy and Uncertainty

In this section, we analyze how the number of Monte Carlo samples $N$ affects predictive accuracy and uncertainty for the in-distribution (ID) OBQA dataset and the out-of-distribution (OoD) ARC dataset at a fixed checkpoint, consistent with the OoD protocol of Table 6 (trained on OBQA; ARC is a shifted evaluation set). For each $N$,

we report accuracy (ACC), negative log-likelihood (NLL), expected calibration error (ECE; 15 bins), and the average per-batch inference time.

*Table 8.* Effect of the number of Monte Carlo samples $N$ on OoD performance on ARC at the same checkpoint. Time denotes average per-batch inference time in seconds.

| $N$ | NLL ↓ | ACC ↑ | ECE ↓ | Time (s) ↓ |
|---|---|---|---|---|
| 1 | 0.4245 | 86.97 | 6.04 | 0.7610 |
| 2 | 0.4245 | 86.62 | 6.31 | 1.5253 |
| 3 | 0.4253 | 86.62 | 6.44 | 2.2888 |
| 4 | 0.4252 | 86.62 | 6.05 | 3.0507 |
| 5 | 0.4252 | 86.80 | 5.84 | 3.8133 |
| 6 | 0.4248 | 86.80 | 5.72 | 4.5761 |
| 7 | 0.4246 | 86.62 | 6.06 | 5.3384 |
| 8 | 0.4246 | 86.80 | 5.88 | 6.1010 |
| 9 | 0.4249 | 86.80 | 5.72 | 6.8632 |
| 10 | 0.4246 | 86.80 | 5.99 | 7.6256 |

*Table 9.* Effect of the number of Monte Carlo samples $N$ on ID performance on OBQA at a fixed checkpoint. Time denotes average per-batch inference time in seconds.

| $N$ | NLL ↓ | ACC ↑ | ECE ↓ | Time (s) ↓ |
|---|---|---|---|---|
| 1 | 1.0756 | 63.33 | 15.55 | 2.3997 |
| 2 | 1.0723 | 63.54 | 14.76 | 4.7858 |
| 3 | 1.0716 | 63.33 | 15.16 | 7.1729 |
| 4 | 1.0707 | 63.54 | 14.80 | 9.5596 |
| 5 | 1.0708 | 63.13 | 15.21 | 11.9460 |
| 6 | 1.0704 | 63.54 | 14.88 | 14.3300 |
| 7 | 1.0699 | 63.33 | 15.03 | 16.7140 |
| 8 | 1.0697 | 63.54 | 14.82 | 19.0961 |
| 9 | 1.0697 | 63.54 | 15.09 | 21.4805 |
| 10 | 1.0702 | 63.54 | 14.91 | 23.8661 |

As shown in Tables 8 and 9, inference time grows approximately linearly with $N$. On the OoD data (ARC), ACC, NLL, and ECE stabilize once $N \geq 2$, with only minor fluctuations beyond that point. On the ID data (OBQA), increasing $N$ yields slightly lower NLL and ECE, consistent with reduced Monte Carlo noise in the predictive distribution. However, the gains beyond N = 2–4 are marginal compared to the additional latency, supporting our practical recommendation of N = 2–4 as a good trade-off between uncertainty quality and computational cost.

## D. Datasets and Preprocessing

This appendix details the six benchmarks used in our evaluation, together with the scoring rules, prompting templates, and implementation choices shared across datasets.

### D.1. Task Overview

All tasks are cast as *closed-set prediction* to avoid generation artifacts. For multiple-choice datasets (ARC-C/E, OBQA) and cloze-style coreference (WinoGrande S/M), we com-

pute option probabilities without free decoding; for BoolQ, we map to a binary verbalizer. Table 10 summarizes formats.

**Common evaluation protocol.** Unless otherwise noted: (i) we use the official training/validation/test splits; (ii) Accuracy (ACC ↑), Expected Calibration Error (ECE ↓, 15 bins on $[0, 1]$ using the model's predicted probability for the chosen label), and Negative Log-Likelihood (NLL ↓) are reported; (iii) probabilities are computed from normalized option log-likelihoods as detailed below; (iv) casing and punctuation are preserved.

**Tokenizer and limits.** We use the backbone's Sentence-Piece tokenizer. Inputs are truncated to a maximum of $L_{\max}$ tokens (default 1024) by trimming long contexts first while keeping all answer options intact. Padding is on the right; the BOS/EOS usage follows the backbone defaults.

**Scoring rule for multiple-choice/cloze.** Given a prefix prompt $x$ and an option string $y_j$ tokenized as $(y_{j,1}, \ldots, y_{j,T_j})$, we score

$$ s_j \;=\; \frac{1}{T_j} \sum_{t=1}^{T_j} \log p_\theta(y_{j,t} \mid x, y_{j,<t}) $$

i.e., *length-normalized* log-likelihood to mitigate option-length bias. Predicted label $\hat{y} = \arg\max_j s_j$; option probabilities are $\pi_j \propto \exp(s_j)$ and are used for ECE/NLL. NLL is $-\log \pi_{y^\star}$ for gold label $y^\star$. In the case of ties, we choose the option with the larger unnormalized (sum) log-likelihood, then lexicographically.

### D.2. Prompting Templates

To minimize prompt sensitivity, we use deterministic templates without instructions or chain-of-thought prompting. Placeholders are written as <...>.

**WinoGrande (S/M).** Each instance contains a sentence with a blank and two candidate fillers.

```
Sentence:  <sentence>
Option A: <sentence with option1>
Option B: <sentence with option2>
```

We compare the completion likelihoods as in the scoring rule.

### ARC-C / ARC-E.

```
Question:  <question>
Options:
A. <choice A>
B. <choice B>
```

```
C. <choice C>
D. <choice D> [E. <choice E> if
present]
Answer:
```

Each option is scored by appending its text after `Answer:`.

**OpenBookQA (OBQA).** Same as ARC; four options (A–D). We omit the provided "facts" in the main results for parity.

**BoolQ.** Binary QA with verbalizers `Yes`/`No`.

```
Passage:  <passage>
Question:  <question>
Answer:
```

We score `Yes` and `No` as the two candidates.

### D.3. Preprocessing and Normalization

- **Unicode/whitespace.** Normalize Unicode quotes/dashes; strip leading/trailing whitespace; collapse repeated spaces inside fields without altering semantics.

- **Deduplication.** Remove exact duplicate training examples (rare in these corpora); evaluation splits remain untouched.

- **Invalid items.** For ARC items with empty or malformed options, we drop the instance *from training only* and keep evaluation intact; such cases are logged ($< 0.1\%$ in our runs).

- **Context truncation.** When sequences exceed $L_{\max}$, we retain the full question and all options and truncate supporting passages from the left (BoolQ) or auxiliary fields (if used), preserving the end of the passage, which often contains the answer evidence.

- **Label integrity.** All label letters (A/B/...) are taken from the official annotations; we do not remap or reorder options.

### D.4. Dataset-specific Notes

**WinoGrande S/M.** We use the official S and M partitions. Following common practice, we evaluate on the validation set for early stopping and report test numbers using the official held-out split when available. No external coreference resources are used.

**ARC-Challenge / ARC-Easy.** We use the AI2 ARC v1 format. Some questions have three or five options; our scoring rule handles variable cardinality. We do not incorporate retrieval or external knowledge in the main comparison.

*Table 10.* Task formats and primary metrics.

| Dataset | Task type | Candidates | Primary metric(s) |
|---|---|---|---|
| WinoGrande-S/M (WG-S/M) | Cloze coreference (2-way) | 2 | ACC, ECE, NLL |
| ARC-Challenge (ARC-C) | Multiple-choice science QA | 3–5 (mostly 4) | ACC, ECE, NLL |
| ARC-Easy (ARC-E) | Multiple-choice science QA | 3–5 (mostly 4) | ACC, ECE, NLL |
| OpenBookQA (OBQA) | Multiple-choice science QA | 4 | ACC, ECE, NLL |
| BoolQ | Yes/No reading comprehension | 2 | ACC, ECE, NLL |

**OpenBookQA.** We use the main OBQA v1.1 multiple-choice set. The "open book" facts are omitted in the main results for parity; adding them can increase accuracy, but does not change the calibration trends observed.

**BoolQ.** We keep case and punctuation in passages. Extremely long passages are truncated from the left to respect $L_{\max}$ while keeping the full question. Verbalizers are fixed as `Yes`/`No`; alternative synonyms (e.g., `True`/`False`) yield similar results.

## E. Metrics for Uncertainty Quantification

NLL and ECE are two common metrics for quantifying model uncertainty. We note that ECE with a fixed number of bins provides a coarse summary of calibration. We use 15-bin ECE throughout for consistency with prior work (Yang et al., 2024; Guo et al., 2017).

**Expected Calibration Error (ECE).** We compute ECE over $B = 15$ equal-width bins on $[0, 1]$ for the predicted confidences $c_i$. Let $I_b = ((b-1)/B, \, b/B]$ and $B_b = \{\, i : c_i \in I_b \,\}$. Then:

$$\text{ECE} = \sum_{b=1}^{B} \frac{|B_b|}{N} \left| \text{acc}(B_b) - \text{conf}(B_b) \right| \quad (41)$$

$$\text{acc}(B_b) = \frac{1}{|B_b|} \sum_{i \in B_b} \mathbf{1}\{\hat{y}_i = y_i\} \quad (42)$$

$$\text{conf}(B_b) = \frac{1}{|B_b|} \sum_{i \in B_b} c_i, \; c_i = \max_k p_\theta(y{=}k \mid x_i) \quad (43)$$

**Negative Log-Likelihood (NLL).** For instance $i$ with gold label $y_i$ and predicted distribution $\hat{\mathbf{p}}_i$,

$$\text{NLL} = -\frac{1}{N} \sum_i \log \hat{p}_{i,y_i} \quad (44)$$

where $\hat{p}_{i,y_i}$ is the probability the model assigns to the gold label $y_i$. For datasets with variable option counts, $\hat{\mathbf{p}}_i$ is always the softmax of length-normalized scores across the *present* options.

## F. Hyperparameters

Hyperparameters are crucial for reproducibility; therefore, we list all hyperparameters used in BAYESIAN-LORA. See Table 11. We also expose these hyperparameters as configurable options to support reproducibility studies.

*Table 11.* Hyperparameters used in our experiments.

| Hyperparameter | Value |
|---|---|
| Optimizer | AdamW |
| Learning rate | $5 \times 10^{-4}$ |
| Betas | (0.9, 0.999) |
| Epsilon ($\epsilon$) | $1 \times 10^{-5}$ |
| Weight decay | 0.1 |
| Scheduler | MultiStepLR (milestones = [4, 6], $\gamma = 0.1$) |
| Epochs | 10 |
| Batch size (train) | 16 (WG-S/M, BoolQ), 8 (ARC-C/E, OBQA) |
| Batch size (eval) | 32 (all datasets) |
| MC samples ($N$) | 4 (commonsense, OoD, and math); 2 (WikiText-2); configurable in cfg.eval |
| KL scaling | 0.2/steps_per_epoch |
| Evaluation frequency | Every 2 epochs |
| GPU | NVIDIA H100 |

The core settings of the Sparse Gaussian Process are listed in Table 12. Inducing rows and columns are set to 9, matching the LoRA rank for a comparable parameter count. The attention Q and K matrices, together with the language model head weight matrices, are replaced by the Bayesian framework (SGP).

## G. Out-of-Distribution Dataset

Following (Yang et al., 2024), we use the Computer Science (CS), Engineering (Eng), Law, and Health subsets of the MMLU dataset as out-of-distribution data to evaluate the robustness of BAYESIAN-LORA. Table 13 summarizes the MMLU subjects used for OoD evaluation. Following the

*Table 12.* Inducing hyperparameters.

| Hyperparameter | Value |
|---|---|
| inducing_rows | 9 |
| inducing_cols | 9 |
| whitened_u | True |
| q_inducing | diagonal |
| learn_lambda | True |
| init_lambda | 0.001 |
| max_lambda | 0.03 |
| max_sd_u | 0.1 |
| cache_cholesky | True |
| prior_sd | 0.1 |
| sqrt_width_scaling | True |
| key_layers | {q_proj, k_proj, lm_head} |

*Table 13.* MMLU subjects and tasks.

| Subject | Tasks |
|---|---|
| Computer Science (CS) | college computer science, computer security, |
| | high school computer science, machine learning |
| Engineering (Eng) | electrical engineering |
| Law | international law, jurisprudence, professional law |
| Health | anatomy, clinical knowledge, college medicine, human aging, |
| | nutrition, professional medicine, virology |

HuggingFace description of the MMLU dataset (Hendrycks et al., 2021), we split college computer science, computer security, high school computer science, and machine learning into the CS category, electrical engineering into the Eng category, international law, jurisprudence, and professional law into the Law category, and anatomy, clinical knowledge, college medicine, human aging, nutrition, professional medicine, and virology into the Health category.

## H. Constrained Bayesian Optimization

In this section, we report the constrained Bayesian optimization results across all datasets. Figure 3 presents the relationship between Negative Log-Likelihood and Expected Calibration Error with the optimal choice explicitly marked.

We additionally present the ARC-Easy Pareto charts in Figure 4 (left: ACC vs. NLL; right: ACC vs. ECE).

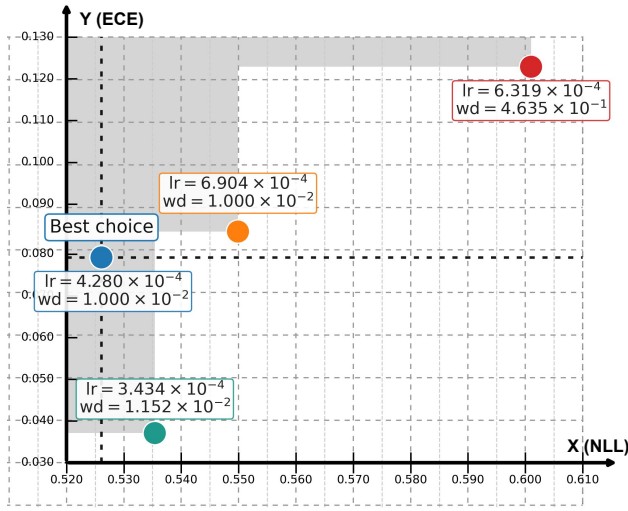

WinoGrande-M (NLL vs ECE)

*Figure 3.* Pareto analysis on the WinoGrande-M dataset (NLL vs. ECE). Each point denotes a hyperparameter pair $(\mathrm{lr}, \mathrm{wd})$. Gray regions show dominated solutions, and the cross marks the "Best choice" near the Pareto front.

Table 15 lists non-dominated candidates (w.r.t. ACC↑, NLL↓, ECE↓) returned by constrained BO for all datasets.

We select the final operating point by proximity to the empirical Pareto front (ties broken by lower NLL).

We tune learning rate $\eta$ and weight decay $\lambda_{\mathrm{wd}}$ on a bounded domain $\mathcal{X} \subset \mathbb{R}^2$:

$$\min_{\mathbf{x} \in \mathcal{X}} \mathbf{f}(\mathbf{x}) = \big(f_1(\mathbf{x}), f_2(\mathbf{x}), f_3(\mathbf{x})\big) = \big(\mathrm{ECE}, \ \mathrm{NLL}, \ -\mathrm{ACC}\big) \tag{45}$$

with inequality constraints

$$c_j(\mathbf{x}) \leq 0, \quad j = 1, \ldots, J. \tag{46}$$

$$y_m(\mathbf{x}) = f_m(\mathbf{x}) + \varepsilon_m, \quad \varepsilon_m \sim \mathcal{N}(0, \sigma_m^2), \\ \tilde{y}_j(\mathbf{x}) = c_j(\mathbf{x}) + \tilde{\varepsilon}_j, \ \tilde{\varepsilon}_j \sim \mathcal{N}(0, \tilde{\sigma}_j^2) \tag{47}$$

For each objective/constraint:

$$f_m \sim \mathcal{GP}\big(\mu_m, k_m\big), \qquad c_j \sim \mathcal{GP}\big(\mu_j^{(c)}, k_j^{(c)}\big) \tag{48}$$

Given data $\mathcal{D}$ and a candidate batch $X = \{\mathbf{x}^{(q)}\}_{q=1}^{Q}$,

$$\mathbf{f}_m(X) \mid \mathcal{D} \sim \mathcal{N}\big(\boldsymbol{\mu}_{m|n}(X), \boldsymbol{\Sigma}_{m|n}(X)\big), \tag{49}$$

$$\boldsymbol{\mu}_{m|n}(X) = \mu_m(X) + K_m(X, X_{1:n}) \\ \times \big(K_m + \sigma_m^2 I\big)^{-1}\big(\mathbf{y}_m - \mu_m(X_{1:n})\big) \tag{50}$$

$$\boldsymbol{\Sigma}_{m|n}(X) = K_m(X, X) - K_m(X, X_{1:n}) \\ \times \big(K_m + \sigma_m^2 I\big)^{-1} K_m(X_{1:n}, X) \tag{51}$$

*Table 14.* Metrics before vs. after constrained Bayesian optimization. "Before" uses Table 1 settings (BAYESIAN-LoRA, N = 4, early stop); "After (BO)" are results under constrained BO.

| | ACC $\uparrow$ | | ECE $\downarrow$ | | NLL $\downarrow$ | | Hyperparams (BO) | |
|---|---|---|---|---|---|---|---|---|
| **Dataset** | Before | After (BO) | Before | After (BO) | Before | After (BO) | LR (BO) | WD (BO) |
| WinoGrande-S | 70.90 | **72.94** | 4.90 | **2.74** | 0.79 | **0.55** | $9.792 \times 10^{-5}$ | $9.339 \times 10^{-2}$ |
| ARC-C | 68.90 | **69.60** | 9.20 | **6.10** | **0.77** | 0.86 | $4.955 \times 10^{-5}$ | $2.056 \times 10^{-1}$ |
| ARC-E | 85.90 | **88.38** | 5.30 | **4.97** | **0.38** | 0.40 | $5.456 \times 10^{-4}$ | $2.707 \times 10^{-2}$ |
| WinoGrande-M | 74.30 | **76.34** | **3.00** | 7.92 | 0.62 | **0.53** | $4.280 \times 10^{-4}$ | $1.000 \times 10^{-2}$ |
| OBQA | 81.60 | **82.80** | **5.70** | 5.84 | **0.49** | 0.54 | $2.135 \times 10^{-4}$ | $2.025 \times 10^{-1}$ |
| BoolQ | 86.10 | **86.41** | **2.10** | 3.10 | 0.29 | **0.28** | $5.421 \times 10^{-4}$ | $3.582 \times 10^{-1}$ |

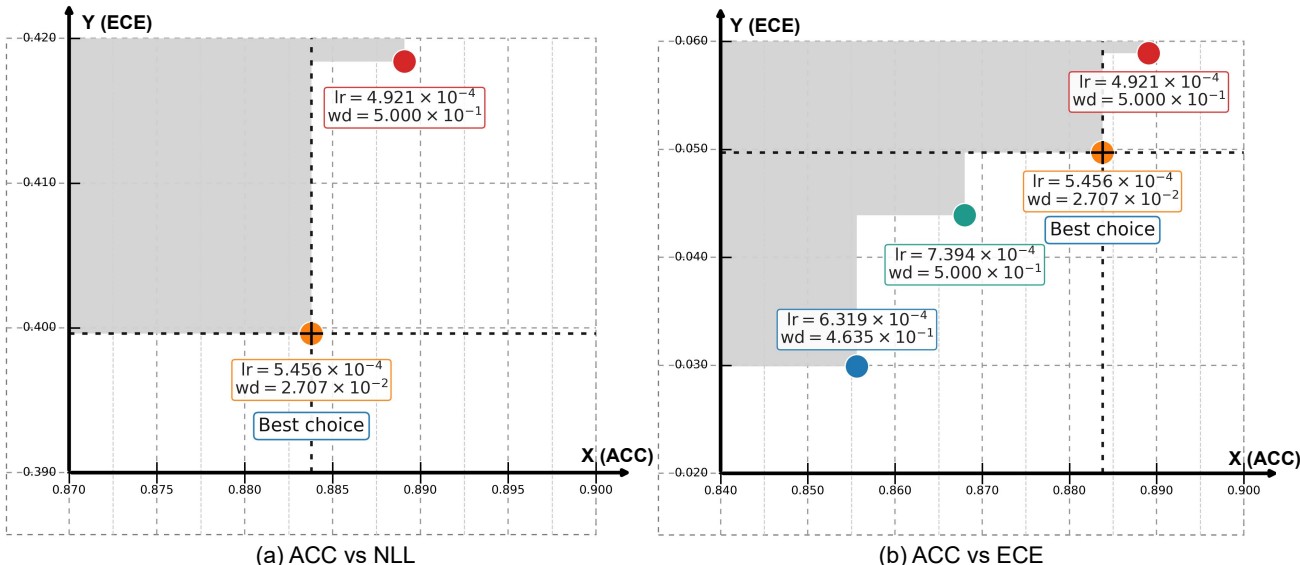

(a) ACC vs NLL  (b) ACC vs ECE

*Figure 4.* Pareto analysis on the ARC-Easy dataset (ACC vs. NLL and ACC vs. ECE). Each point denotes a hyperparameter pair (lr, wd). Gray regions show dominated solutions, and the cross marks the "Best choice" near the Pareto front.

and analogously for constraints.

With independent constraints,

$$\text{PoF}(X) \approx \prod_{q=1}^{Q} \prod_{j=1}^{J} \Phi\left( \frac{-\mu_{j|n}^{(c)}(\mathbf{x}^{(q)})}{\sqrt{\Sigma_{j|n}^{(c)}(\mathbf{x}^{(q)}, \mathbf{x}^{(q)})}} \right) \quad (52)$$

where $\Phi$ is the standard normal CDF.

**q-NEHVI acquisition (constrained).** Let $\mathbf{r}$ be the reference point and $\text{HV}(\cdot; \mathbf{r})$ the dominated hypervolume. Define

$$\alpha_{\text{q-NEHVI}}(X) = \mathbb{E}\Big[ \big(\text{HV}(\tilde{\mathcal{P}} \cup \mathbf{F}(X); \mathbf{r}) - \text{HV}(\tilde{\mathcal{P}}; \mathbf{r})\big)_+,$$

$$\mathbb{I}\{\mathbf{C}(X) \leq \mathbf{0}\} \mid \mathcal{D} \Big] \quad (53)$$

where $\tilde{\mathcal{P}}$ is a sample of the posterior Pareto set (feasible and non-dominated), $\mathbf{F}(X)$ stacks the objective draws, and $\mathbf{C}(X)$ the constraint draws.

With Cholesky factors $\mathbf{L}_{m|n}(X)$ of (51),

$$\mathbf{F}_m^{(t)}(X) = \boldsymbol{\mu}_{m|n}(X) + \mathbf{L}_{m|n}(X)\mathbf{z}_m^{(t)},$$

$$\mathbf{z}_m^{(t)} \sim \mathcal{N}(\mathbf{0}, I) \quad (54)$$

and similarly $\mathbf{C}^{(t)}(X)$. Sampling $\tilde{\mathcal{P}}^{(t)}$ from the posterior of historical designs, the MC estimator is

$$\widehat{\alpha}_{\text{q-NEHVI}}(X) = \frac{1}{T} \sum_{t=1}^{T} \Big( \text{HV}(\tilde{\mathcal{P}}^{(t)} \cup \mathbf{F}^{(t)}(X); \mathbf{r})$$

$$- \text{HV}(\tilde{\mathcal{P}}^{(t)}; \mathbf{r}) \Big)_+ \mathbb{I}\{\mathbf{C}^{(t)}(X) \leq \mathbf{0}\} \quad (55)$$

A PoF-weighted variant:

$$\widehat{\alpha}_{\text{q-NEHVI}}^{(\text{PoF})}(X) = \left[ \frac{1}{T} \sum_{t=1}^{T} \Big( \text{HV}(\tilde{\mathcal{P}}^{(t)} \cup \mathbf{F}^{(t)}(X); \mathbf{r}) \right.$$

$$\left. - \text{HV}(\tilde{\mathcal{P}}^{(t)}; \mathbf{r}) \Big)_+ \right] \quad (56)$$

$$\times \text{PoF}(X)$$

*Table 15.* Full Pareto candidates after constrained BO (all datasets combined).

| Dataset | Cand. | Acc | NLL | ECE (%) | LR | Weight Decay |
|---|---|---|---|---|---|---|
| WinoGrande-M | 1 | 0.7753 | 0.6010 | 12.29 | $6.319 \times 10^{-4}$ | $4.635 \times 10^{-1}$ |
| WinoGrande-M | 2 | 0.7682 | 0.5499 | 8.34 | $6.904 \times 10^{-4}$ | $1.000 \times 10^{-2}$ |
| WinoGrande-M | 3 | 0.7634 | 0.5260 | 7.92 | $4.280 \times 10^{-4}$ | $1.000 \times 10^{-2}$ |
| WinoGrande-M | 4 | 0.7342 | 0.5354 | 3.71 | $3.434 \times 10^{-4}$ | $1.152 \times 10^{-2}$ |
| WinoGrande-S | 1 | 0.7342 | 0.5870 | 9.90 | $7.832 \times 10^{-5}$ | $8.091 \times 10^{-2}$ |
| WinoGrande-S | 2 | 0.7294 | 0.5474 | 2.74 | $9.792 \times 10^{-5}$ | $9.339 \times 10^{-2}$ |
| WinoGrande-S | 3 | 0.7033 | 0.5740 | 2.42 | $8.810 \times 10^{-5}$ | $7.783 \times 10^{-2}$ |
| ARC-C | 1 | 0.6959 | 0.8616 | 6.10 | $4.955 \times 10^{-5}$ | $2.056 \times 10^{-1}$ |
| ARC-C | 2 | 0.6858 | 0.8556 | 9.16 | $9.581 \times 10^{-5}$ | $5.000 \times 10^{-1}$ |
| ARC-C | 3 | 0.6014 | 1.0015 | 4.41 | $6.081 \times 10^{-5}$ | $3.141 \times 10^{-1}$ |
| ARC-E | 1 | 0.8891 | 0.4184 | 5.89 | $4.921 \times 10^{-4}$ | $5.000 \times 10^{-1}$ |
| ARC-E | 2 | 0.8838 | 0.3996 | 4.97 | $5.456 \times 10^{-4}$ | $2.707 \times 10^{-2}$ |
| ARC-E | 3 | 0.8680 | 0.4495 | 4.39 | $7.394 \times 10^{-4}$ | $5.000 \times 10^{-1}$ |
| ARC-E | 4 | 0.8556 | 0.4797 | 2.99 | $6.319 \times 10^{-4}$ | $4.635 \times 10^{-1}$ |
| OBQA | 1 | 0.8380 | 0.5499 | 7.18 | $1.180 \times 10^{-3}$ | $4.635 \times 10^{-1}$ |
| OBQA | 2 | 0.8280 | 0.5411 | 5.84 | $2.135 \times 10^{-4}$ | $2.025 \times 10^{-1}$ |
| BoolQ | 1 | 0.8641 | 0.2841 | 3.10 | $5.421 \times 10^{-4}$ | $3.582 \times 10^{-1}$ |
| BoolQ | 2 | 0.8572 | 0.3025 | 2.54 | $4.987 \times 10^{-4}$ | $2.041 \times 10^{-1}$ |
| BoolQ | 3 | 0.8490 | 0.3152 | 1.86 | $6.103 \times 10^{-4}$ | $5.000 \times 10^{-1}$ |

The reference point is then obtained as follows:

$$\mathbf{r} = (r_1, r_2, r_3), \qquad r_m < \min_{\text{historical feasible}} f_m \qquad (57)$$

*Table 16.* Trainable parameters corresponding to different ranks.

| Rank | 4 | 9 | 16 | 32 | 64 | 128 |
|---|---|---|---|---|---|---|
| **Trainable Parameters** | 3.4M | 4.9M | 7.0M | 12M | 22M | 43M |

## I. Additional Experiments on Qwen3-14B

To further verify that the benefits of BAYESIAN-LoRA extend to more recent backbones, we conduct additional experiments on **Qwen3-14B**, a newer-generation dense 14B model that is distinct from the Qwen2.5-14B-Instruct backbone used for the MATH experiments in Section 5.2; the two are different models evaluated on separate tasks. These Qwen3-14B experiments are reported only in this appendix and are not part of the main-paper results, which foreground Qwen2.5-14B-Instruct and Qwen3-30B-A3B-Instruct-2507. We evaluate on the multiple-choice classi-

fication benchmarks ARC-Challenge (ARC-C) and Open-BookQA (OBQA), as well as on the mathematical reasoning benchmark AIME 2024. The number of Monte Carlo samples is set to N = 4. The classification results are summarized in Table 17, and the AIME 2024 results in Table 18.

On ARC-C and OBQA, BAYESIAN-LoRA matches or exceeds all LoRA-style baselines in accuracy while substantially improving calibration: ECE drops from 4.30 (LoRA) to 3.30 on ARC-C and from 5.04 to 3.40 on OBQA, and NLL improves by a similar margin. These results are consistent with the Llama-2-7B trends reported in the main paper and show that the structured Kronecker-factored posterior continues to deliver well-calibrated predictions when scaled to a more recent 14B-parameter backbone. A matched-regularization comparison against LoRA with MC-Dropout (Gal & Ghahramani, 2016)—which injects stochasticity without any Bayesian formulation—is provided on Llama-2-7B in Table 1, where BAYESIAN-LoRA achieves consistently lower ECE and NLL across all six benchmarks; this isolates the calibration gain to the structured posterior rather than to noise injection alone.

*Table 17.* Qwen3-14B on classification benchmarks N = 4. ACC (↑), ECE (↓), NLL (↓). Best results in each column are in **bold**.

| Method | ARC-C | | | OBQA | | |
|---|---|---|---|---|---|---|
| | ACC ↑ | ECE ↓ | NLL ↓ | ACC ↑ | ECE ↓ | NLL ↓ |
| LoRA | 90.70 | 4.30 | 0.4259 | 88.80 | 5.04 | 0.3942 |
| LoRA+ | 90.71 | 4.12 | 0.4494 | 89.20 | 4.43 | 0.3855 |
| AdaLoRA | 90.61 | 4.48 | 0.5962 | 89.80 | 4.77 | 0.5087 |
| Bayesian-LoRA | **90.78** | **3.30** | **0.2681** | **90.40** | **3.40** | **0.3137** |

*Table 18.* Qwen3-14B on AIME 2024 (N = 4; train: AIME 1983–2023, test: 30 problems). We report accuracy (ACC, %) together with ECE and NLL of the chain-of-thought predictive distribution. Best results are in **bold**.

| Method | ACC ↑ | ECE ↓ | NLL ↓ |
|---|---|---|---|
| Qwen3-14B base (4-shot) | 76.7 | 9.07 | 0.2856 |
| Standard LoRA | **80.0** | 11.26 | 0.2982 |
| Bayesian-LoRA (epoch 5) | **80.0** | **5.26** | **0.1144** |

Overall, these additional Qwen3-14B experiments corroborate our main findings: BAYESIAN-LoRA provides consistent gains in calibration and likelihood across both classification and mathematical reasoning tasks, and the improvements are not specific to a single backbone among those used in the main experiments (Llama-2-7B, Qwen2.5-14B-Instruct, and Qwen3-30B-A3B).

## J. Comparison with Recent Bayesian LoRA Baselines

To position BAYESIAN-LoRA against the most recent uncertainty-aware LoRA methods, we compare against two concurrent works: **C-LoRA** (Rahmati et al., 2025), which models uncertainty through contextual low-rank adaptation, and **TFB** (Shi et al., 2025), a post-hoc, training-free Bayesianization procedure applied on top of an existing LoRA adapter. These approaches were published after the initial ICML submission deadline; we include them here for completeness.

**Comparison with C-LoRA on Llama-2-7B.** C-LoRA uses the same backbone as our main experiments, allowing a direct comparison on all six commonsense reasoning benchmarks. Table 19 reports ACC, ECE, and NLL for both methods.

BAYESIAN-LoRA outperforms C-LoRA on accuracy across all 6 benchmarks (with a gain of +1.1 on ARC-C and a maximum gain of +4.69 on WinoGrande-S), on ECE in 2 of 6 benchmarks, and on NLL in 4 of 6 benchmarks.

**Comparison with TFB on Llama-3.1-8B.** TFB uses Llama-3.1-8B as its backbone in the original paper, so we additionally ran BAYESIAN-LoRA on the same backbone to enable a matched comparison. TFB is a post-hoc procedure applied on top of a BLoB-Mean checkpoint; we report the numbers directly from Table 1 of (Shi et al., 2025). Results are in Table 20.

On Llama-3.1-8B, the two methods reach very similar accuracy (within 0.7 points on both datasets), with TFB marginally higher; however, BAYESIAN-LoRA achieves substantially better calibration: ECE drops by roughly 50% on ARC-C (6.48 → 3.23) and 25% on ARC-E (2.44 → 1.83), and NLL is lower on both datasets. This is consistent with the conceptual difference between the two approaches: TFB applies a training-free Bayesian transformation after standard fine-tuning, whereas BAYESIAN-LoRA optimizes the ELBO end-to-end, so calibration shapes the posterior throughout training rather than being added as a post-hoc correction.

**Summary.** We view C-LoRA, TFB, and BAYESIAN-LoRA as complementary contributions that approach uncertainty-aware LoRA from different angles—contextual conditioning, training-free post-hoc Bayesianization, and end-to-end variational inference with a GP-inspired posterior, respectively. Collectively, the comparisons in Tables 19–20 indicate that BAYESIAN-LoRA delivers competitive accuracy with consistently stronger calibration on most of the benchmarks we evaluated.

## K. Visual Comparison with Bayesian and Post-hoc PEFT Baselines

To make the positioning of BAYESIAN-LoRA relative to existing Bayesian and post-hoc calibration methods more intuitive, Figures 5 and 6 plot Accuracy against Expected Calibration Error for all baselines on Llama-2-7B. In both plots the ECE axis is reversed (low ECE on the right), so the upper-right corner corresponds to the ideal regime—high accuracy with low calibration error—and a method that dominates another lies strictly closer to that corner.

Figure 5 summarizes the average performance across the

*Table 19.* Comparison with C-LoRA (Rahmati et al., 2025) on Llama-2-7B across six commonsense reasoning benchmarks. BAYESIAN-LORA uses N = 4 MC samples; C-LoRA results are reported with $M = 10$. Best results in each row are in **bold**.

| Metric | Method | WinoGrande-S | ARC-C | ARC-E | WinoGrande-M | OBQA | BoolQ |
|---|---|---|---|---|---|---|---|
| ACC ↑ | C-LoRA ($M = 10$) | 66.21 | 67.79 | 84.38 | 70.48 | 78.26 | 84.64 |
| | BAYESIAN-LORA (N = 4) | **70.90** | **68.90** | **85.90** | **74.30** | **81.60** | **86.10** |
| ECE ↓ | C-LoRA ($M = 10$) | 6.86 | **8.83** | **4.27** | 3.71 | **4.00** | **1.62** |
| | BAYESIAN-LORA (N = 4) | **4.90** | 9.20 | 5.30 | **3.00** | 5.70 | 2.10 |
| NLL ↓ | C-LoRA ($M = 10$) | **0.63** | 0.88 | 0.48 | **0.57** | 0.59 | 0.35 |
| | BAYESIAN-LORA (N = 4) | 0.79 | **0.77** | **0.38** | 0.62 | **0.49** | **0.29** |

*Table 20.* Comparison with TFB (Shi et al., 2025) on Llama-3.1-8B. TFB numbers are taken from Table 1 of (Shi et al., 2025). Best results in each row are in **bold**.

| Metric | Method | ARC-C | ARC-E |
|---|---|---|---|
| ACC ↑ | BLoB-Mean + TFB | $83.33 \pm 0.19$ | $91.76 \pm 0.48$ |
| | BAYESIAN-LORA | 83.01 | 91.12 |
| ECE ↓ | BLoB-Mean + TFB | $6.48 \pm 0.36$ | $2.44 \pm 0.50$ |
| | BAYESIAN-LORA | **3.23** | **1.83** |
| NLL ↓ | BLoB-Mean + TFB | $0.530 \pm 0.04$ | $0.230 \pm 0.02$ |
| | BAYESIAN-LORA | **0.503** | **0.217** |

six commonsense reasoning benchmarks (WinoGrande-S, WinoGrande-M, ARC-C, ARC-E, OBQA, BoolQ). BAYESIAN-LORA (red star) lies on the upper-right Pareto frontier, jointly achieving the highest accuracy and the lowest ECE. Training-time Bayesian baselines—BLoB (Wang et al., 2024b), MC Dropout (Gal & Ghahramani, 2016), checkpoint ensembles, and BBB (Blundell et al., 2015)—trade off either accuracy or calibration; post-hoc calibration methods (Temperature Scaling (Guo et al., 2017), Laplace Approximation (Daxberger et al., 2021; Yang et al., 2024)) reduce ECE but do not improve accuracy and remain below BAYESIAN-LORA on both axes.

Figure 6 shows the same comparison restricted to the WinoGrande-S benchmark, which is the smallest and most prone to overconfidence after fine-tuning. BAYESIAN-LORA again sits in the upper-right region, while BBB collapses to near-chance accuracy (consistent with the high variance reported in the main paper), and post-hoc methods reduce ECE only at the cost of lower accuracy relative to BAYESIAN-LORA.

Together, these two figures provide a visual summary of the quantitative comparisons in Table 1 and complement the conceptual contrast between three families of methods: post-hoc methods operate on a fixed point estimate after training, BBB-style approaches place a mean-field Gaussian over every LoRA parameter, while BAYESIAN-LORA places a structured GP-inspired posterior over a low-dimensional

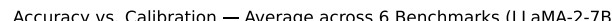

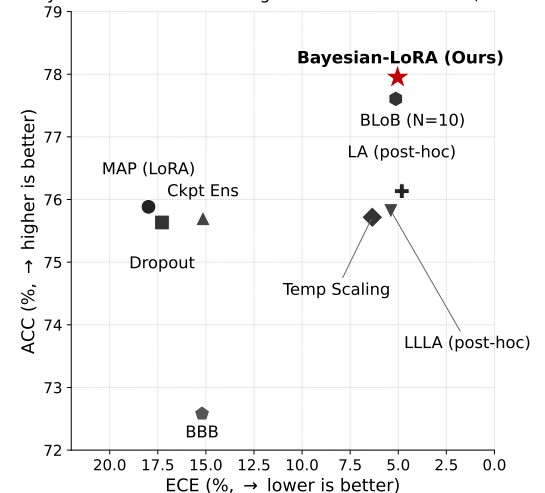

*Figure 5.* Accuracy (ACC, ↑) vs. Expected Calibration Error (ECE, ↓) on Llama-2-7B, averaged across six commonsense reasoning benchmarks. Each marker is a method. BAYESIAN-LORA (red star) occupies the upper-right Pareto-optimal corner, outperforming both training-time Bayesian baselines (BLoB, Dropout, checkpoint ensembles, BBB) and post-hoc calibration baselines (Temperature Scaling, Laplace, Last-Layer Laplace). The ECE axis is reversed (low ECE on the right), so the ideal region, with high ACC and low ECE, is the upper-right corner.

inducing matrix and optimizes the ELBO end-to-end. The resulting joint position in (ACC, ECE) space reflects this design choice.

## L. Stability of BAYESIAN-LORA: Seeds, Adapter Placement, and Inducing-Matrix Shape

To show that the reported gains of BAYESIAN-LORA are not driven by a particular random seed, a particular choice of adapter modules, or a particular shape of the inducing matrix, we conduct three robustness studies on ARC-Easy. All runs in this section use N = 4 Monte Carlo samples on Qwen3-14B.

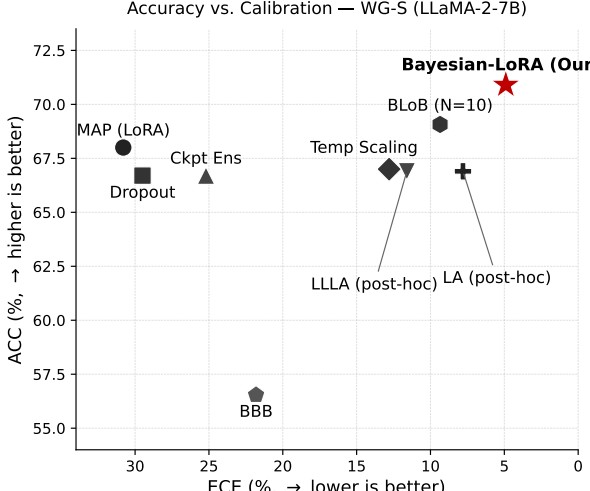

*Figure 6.* Accuracy (ACC, ↑) vs. Expected Calibration Error (ECE, ↓) on the WinoGrande-S (WG-S) benchmark with Llama-2-7B. BAYESIAN-LORA (red star) lies on the upper-right Pareto frontier, achieving the highest accuracy while maintaining competitive calibration. BBB collapses to near-chance accuracy on this small binary task, and post-hoc methods (Temperature Scaling, Laplace, Last-Layer Laplace) reduce ECE only at the cost of lower accuracy.

**Seed stability.** Table 21 reports ACC, ECE, and NLL across five random seeds {42, 123, 456, 789, 2024} with all other settings fixed to the default configuration of Table 1. The standard deviation of accuracy is 0.01 points, of ECE 0.06, and of NLL 0.0001, indicating that the calibration gains over LoRA-style baselines are statistically meaningful and do not depend on seed choice.

*Table 21.* Seed stability of BAYESIAN-LORA on ARC-Easy (Qwen3-14B, N = 4). The last row reports mean ± standard deviation over five seeds.

| Seed | ACC ↑ | ECE ↓ | NLL ↓ |
|---|---|---|---|
| 42 | 94.59 | 0.76 | 0.1347 |
| 123 | 94.57 | 0.88 | 0.1345 |
| 456 | 94.57 | 0.88 | 0.1348 |
| 789 | 94.57 | 0.81 | 0.1349 |
| 2024 | 94.57 | 0.76 | 0.1347 |
| **mean ± std** | **94.57 ± 0.01** | **0.82 ± 0.06** | **0.1347 ± 0.0001** |

**Adapter placement.** Table 22 varies which modules receive BAYESIAN-LORA adapters, ranging from the default attention slice (q, k, lm_head) to LoRA-style (q, v, lm_head), full attention (q, k, v, o, lm_head), MLP-only (gate, up, down, lm_head), and all linear layers. Accuracy stays at $\sim 94.5\%$ and ECE in $[0.71, 0.93]$ across all five placements, showing that calibration gains are not specific to the default attention configuration used in the main experiments.

*Table 22.* Adapter placement ablation on ARC-Easy (Qwen3-14B, seed = 42, N = 4). Rows differ only in the set of target modules to which BAYESIAN-LORA adapters are attached.

| Placement | Target modules | ACC ↑ | ECE ↓ | NLL ↓ |
|---|---|---|---|---|
| A (default) | q, k, lm_head | 94.59 | 0.76 | 0.1347 |
| B (LoRA-style) | q, v, lm_head | 94.57 | 0.74 | 0.1347 |
| C (Attention) | q, k, v, o, lm_head | 94.57 | 0.71 | 0.1348 |
| D (MLP) | gate, up, down, lm_head | 94.49 | 0.93 | 0.1348 |
| E (All) | all linear + lm_head | 94.53 | 0.74 | 0.1348 |

**Inducing-matrix shape ($r \neq c$).** Throughout the main paper we set the inducing dimensions $r = c$ to match the LoRA rank for a fair parameter-count comparison. To confirm that this is a convenience rather than a requirement, Table 23 sweeps $r$ and $c$ independently on Qwen3-14B. Performance is essentially flat across symmetric, mildly asymmetric, and strongly asymmetric configurations, with ACC differing by at most 0.06 points and ECE by at most 0.11.

*Table 23.* Ablation over the inducing-matrix shape $(r, c)$ on ARC-Easy (Qwen3-14B, seed = 42, N = 4). Performance is robust to the choice of $r$ versus $c$ and does not require $r = c$.

| Configuration | $r$ | $c$ | ACC ↑ | ECE ↓ | NLL ↓ |
|---|---|---|---|---|---|
| $r = c$ (default) | 9 | 9 | 94.59 | 0.76 | 0.1347 |
| $r < c$ | 4 | 9 | 94.57 | 0.80 | 0.1347 |
| $r > c$ | 9 | 4 | 94.57 | 0.72 | 0.1349 |
| $r < c$ (larger) | 9 | 16 | 94.53 | 0.82 | 0.1349 |
| $r > c$ (larger) | 16 | 9 | 94.53 | 0.83 | 0.1349 |

Collectively, Tables 21–23 indicate that the gains reported in the main paper are stable across random initialization, adapter placement, and inducing-matrix shape, and are not artifacts of a single hyperparameter configuration.

## M. Training Hyperparameters on the MATH Dataset

For completeness, Table 24 lists the full training configuration used for the MATH experiments in Section 5.2, covering both deterministic LoRA and BAYESIAN-LORA fine-tuning. BAYESIAN-LORA uses a structured Bayesian low-rank update with inducing rank $r = c = 9$ (the MATH LoRA baseline uses rank 8).

*Table 24.* Training hyperparameters for deterministic LoRA and BAYESIAN-LORA fine-tuning on the MATH dataset.

| Category | Hyperparameter (value) |
|---|---|
| *Model & tokenizer* | |
| Model name | `Qwen/Qwen3-30B-A3B-Instruct-2507` |
| Max input length | 1024 |
| BF16 precision | True |
| FP16 precision | False |
| Load in 8-bit | False |
| Gradient checkpointing | Optional (default: disabled) |
| Pad token | EOS token |
| *Training* | |
| Epochs | 1 |
| Batch size (per device) | 1 |
| Gradient accumulation steps | 1 |
| Learning rate | $1 \times 10^{-5}$ |
| Warmup steps | 100 |
| Optimizer | AdamW (PyTorch) |
| Max grad norm | 1.0 |
| Logging steps | 50 |
| Save strategy | Per epoch |
| Save total limit | 4 |
| *Dataset & prompt* | |
| Dataset | `DigitalLearningGmbH/MATH-lighteval` |
| Split | train |
| Prompt type | qwen25-math-cot |
| Apply chat template | True |
| Max tokens per call | 1534 |
| *Deterministic LoRA* | |
| LoRA rank $r$ | 8 |
| LoRA $\alpha$ | 16 |
| LoRA dropout | 0.05 |
| Target modules | q_proj, v_proj |
| Bias | none |
| Task type | Causal LM |
| BAYESIAN-LORA | |
| BAYESIAN-LORA layer | `BayesianLoRALayer` |
| Bayesian rank ($r_{\text{bayes}}$) | 9 |
| Linear layer override | `nn.Linear` $\rightarrow$ `BayesianLoRALayer` |
| Posterior treatment | Variational / structured low-rank |
| Uncertainty sampling | Enabled (Bayesian forward samples) |
| Merge for inference | Supported via `merge_and_unload` |
| *Evaluation* | |
| Temperature | 0 |
| Top-p | 1.0 |
| Number of samples | 1 |
| vLLM eval | Optional |
| Eval batch size | 1 |

