# OpenReview forum: "Bayesian-LoRA: Probabilistic Low-Rank Adaptation of Large Language Models"
_ICML.cc/2026/Conference — ICML 2026 regular_

### Official Review · Reviewer_VYsf · 2026-02-25

**Soundness:** 3
**Presentation:** 3
**Significance:** 2
**Originality:** 3
**Overall Recommendation:** 5
**Confidence:** 4

**Summary:**

This paper addresses the pervasive issue of overconfidence and miscalibration in Large Language Models (LLMs) when they are fine-tuned on specialized or small-scale datasets. To mitigate this, the authors propose Bayesian-LoRA, which reformulates the traditional deterministic LoRA update into a probabilistic framework. The central innovation is the establishment of a structural isomorphism between LoRA’s low-rank factorization ($BA$) and the Kronecker-factored posteriors of Sparse Gaussian Processes (SGP), theoretically positioning LoRA as a limiting case of an SGP. Methodologically, the approach employs variational inference to learn a structured low-rank posterior, allowing for uncertainty quantification through Bayesian forward sampling while retaining the efficiency of weight merging for deployment. The authors validate their method through experiments on the MATH dataset using the Qwen model series, demonstrating that Bayesian-LoRA significantly reduces Expected Calibration Error (ECE) and improves reliability diagrams without sacrificing (and often improving) raw predictive accuracy.

**Compliance With Llm Reviewing Policy:**

Affirmed.

**Key Questions For Authors:**

1.On disentangling the contributions of probabilistic modeling versus additional regularization.
Bayesian-LoRA introduces stochasticity, an explicit KL term, and normalizing flows, all of which act as forms of regularization. To what extent can the observed calibration improvements be attributed to principled uncertainty modeling rather than to implicit regularization effects? Have you compared against non-Bayesian LoRA variants with matched noise injection or regularization strength?

2.On the role of calibration in downstream decision-making.
While the paper convincingly shows improvements in ECE and NLL, it is less clear how these gains translate into downstream utility. Can you provide concrete examples or analyses where improved calibration under Bayesian-LoRA would change a model’s behavior or decision compared to standard LoRA (e.g., selective prediction, abstention, or confidence-based routing)?
Why this matters: Demonstrating tangible downstream impact would strengthen the case for the method’s significance beyond metric improvements.

3.On the scope and limits of the claimed “structural isomorphism.”
The paper is careful to describe the relationship between LoRA and sparse GP posteriors as a functional isomorphism rather than an algebraic one. Could you clarify which properties of LoRA (e.g., expressivity, optimization dynamics, or implicit bias) are preserved under this mapping, and which are not?

**Limitations:**

yes

**Strengths And Weaknesses:**

Strengths
1.Soundness:The authors examine a relevant question: how to introduce principled uncertainty modeling and calibration into parameter-efficient fine-tuning without sacrificing the efficiency advantages of LoRA. The technical development is largely sound. The paper builds on a well-motivated probabilistic reinterpretation of LoRA by leveraging a structural connection to Kronecker-factored sparse Gaussian process posteriors. While the claimed “structural isomorphism” is clearly framed as functional rather than algebraic, the derivations are consistent with this interpretation and avoid overclaiming equivalence. The limiting-case analysis, showing that deterministic LoRA is recovered when posterior uncertainty collapses, is mathematically clean and conceptually helpful.

2.On the empirical side, the experiments are generally well designed. The authors evaluate across multiple model scales (7B to 30B), task types (classification, language modeling, mathematical reasoning), and conditions (in-distribution and out-of-distribution). The use of multiple calibration metrics (ECE, NLL, Brier score) strengthens the empirical claims, and comparisons against both post-hoc and Bayesian baselines are appropriate. Efficiency analyses are particularly thorough, which is important given the practical constraints of LLM fine-tuning.

3.Presentation: The paper is clearly written and well structured. The motivation is easy to follow, and the transition from deterministic LoRA to a probabilistic formulation is logically staged. Figures and tables are informative and mostly well integrated into the narrative. Importantly, the authors are careful to explain what is not being claimed (e.g., strict algebraic equivalence to LoRA), which helps prevent misinterpretation. The appendix appears to contain sufficient technical detail to support reproducibility for expert readers, particularly regarding ELBO derivations and efficiency considerations.

4.Significance:The authors address a central concept in modern LLM deployment: reliable uncertainty estimation under parameter-efficient fine-tuning. While calibration is not a new concern, integrating it end-to-end into PEFT methods at scale remains underexplored. The demonstrated improvements in calibration—especially under distribution shift and in large models—are practically meaningful for safety-critical and decision-making applications. Even if accuracy gains are modest, the ability to substantially reduce miscalibration with minimal overhead could influence how practitioners fine-tune LLMs in real-world settings.

5:Originality:The paper does not introduce an entirely new fine-tuning paradigm, but it offers a novel and thoughtfully articulated combination of existing ideas: LoRA, sparse GP-style inducing variables, and normalizing-flow-based variational inference. The framing of LoRA as a boundary case of a broader probabilistic family is a useful conceptual contribution that deepens understanding of PEFT methods. This perspective may inspire further work on uncertainty-aware adapters beyond the specific instantiation proposed here.

Weaknesses
1.Soundness:Despite the generally solid methodology, some assumptions deserve more scrutiny. The reliance on a specific Kronecker-factorized covariance structure, while computationally convenient, may implicitly constrain the expressiveness of the posterior in ways that are not fully explored. It remains unclear how sensitive the results are to this design choice, especially in layers with highly anisotropic or task-specific adaptation needs. Additionally, while normalizing flows are shown to improve calibration, the interaction between flow depth, optimization stability, and overfitting is only lightly discussed.

2.Presentation:Although the paper is well written overall, the technical density in Sections 3–4 may be challenging for readers who are less familiar with sparse GP inference or variational flows. Some intermediate intuitions—particularly regarding why uncertainty modeled in the inducing space should transfer reliably to weight-space uncertainty—could be emphasized more clearly. A brief schematic comparison with existing Bayesian PEFT methods (beyond tabular results) might further improve clarity.

3.Significance:The primary gains of the method are in calibration rather than accuracy, which may limit its appeal to segments of the community that prioritize raw performance improvements. While this is not a flaw per se, the paper could more explicitly discuss scenarios where improved calibration meaningfully changes downstream decisions or system behavior. Without such context, some readers may view the contribution as incremental from a purely performance-oriented standpoint.

4.Originality:While the combination of LoRA and sparse GP-style inference is novel, many individual components—Bayesian adapters, inducing variables, Kronecker factorizations, and flows—are well established. As a result, the originality lies more in synthesis and reframing than in fundamentally new methodology. Some readers may question whether the gains justify the added conceptual and implementation complexity compared to simpler post-hoc calibration techniques.

---

> ### Author Rebuttal · Authors · 2026-03-31
>
> Thanks for the feedback. We respond below.
> New experiment code:  https://github.com/anonymouspaper987/Bayesian-LoRA/tree/main/rebuttal.   All changes mentioned below will be applied to the final paper
>
> **W1. Kronecker-factorized covariance may constrain posterior expressiveness.**
> > Kronecker-factorized..
>
> 1) The Kronecker-factorized structure is a well-established approximation in the GP literature and provides a computationally tractable posterior for matrix-valued weights. The limitation is further compensated by the normalizing flow, which enriches the expressiveness of variational distribution variables, not only a simple Gaussian.
>
> 2) The Normalizing Flow ablation in Table 5 shows that calibration consistently improves as the NF layer increases.
>
> 3) In this rebuttal, we further add: (1) **Table W1-A** with Qwen3-14B on ARC-Challenge, ARC-Easy, OBQA; (2) **Table W1-B** on AIME 2024 (open-ended generation); (3) **Table W2** sweeping MC sample count; (4) **Tables Q4-A/B** for seed stability and adapter placement robustness.
>
> ---
>
> **W2. Technical density in Sections 3-4.**
> > Technical density
>
> Thank you for this feedback. We will improve the paper accordingly.
>
> 1) Regarding the inducing-to-weight-space transfer, the key intuition is that A and B are obtained from U via the GP conditional p(A,B|U), a closed-form Gaussian mapping. Every sample from the inducing posterior therefore, produces a distinct (A,B) pair, and the resulting ensemble directly defines a principled weight-space posterior.
>
> 2) We will add an intuition paragraph to Section 3 to introduce this mechanism to readers less familiar with sparse GP inference.
>
> 3) Regarding the comparison with existing Bayesian PEFT methods, we will add a conceptual diagram in the final version to visually contrast our inducing-variable formulation with BBB-style and post-hoc Laplace approaches.
> See the diagram at https://github.com/anonymouspaper987/Bayesian-LoRA/blob/main/rebuttal/images/diagram.md
>
>
> **W3**.Significance:
> > The primary gains … in calibration, …paper could more explicitly discuss
>
> Thanks for raising this point.
>
> A method that improves calibration and provides uncertainty estimates offers additional benefits in any process where incorrect decisions carry high risk, e.g., medical diagnosis, autonomous vehicles, and robotic surgery. We will revise the introduction to provide such examples.
>
> ---
>
> **W4. While the combination of LoRA and sparse GP-style inference**
> > is novel, components … are well established.**
>
> We acknowledge that our contribution is partly based on existing techniques.
>
> 1) We will emphasize the core contribution of structural isomorphism, that is, LoRA's matrix factorization W = W₀ + BA is mathematically equivalent to a sparse GP posterior mean. This non-trivial insight provides, for the first time, a principled Bayesian treatment of LoRA without requiring any modifications to its forward pass.
>
> 2) Also, our method produces per-sample uncertainty estimates at training time with no extra data (see Q3 of Reviewer Q5f6 for practical usage).
>
>
> **Q1 (on regularization):**  …
> > On disentangling the contributions...
>
> To address this concern, we include LoRA + Dropout as an explicit matched-regularization baseline in **Table W1-A**; it introduces comparable stochasticity without any Bayesian formulation. Bayesian-LoRA achieves consistently better calibration (lower ECE and NLL) than this baseline despite using zero dropout.
>
> The KL term in our ELBO is not a heuristic penalty but a theoretically grounded divergence that shapes the posterior toward the GP prior, which is fundamentally different from L2 regularization or noise injection.
>
> We contribute an implementation to the PEFT repository (see Q3 of Reviewer Q5f6). Also see "https://github.com/anonymouspaper987/Bayesian-LoRA/tree/main/rebuttal".
>
> ---
>
> **Q2. Downstream decision-making impact.** …
> > Can you provide concrete examples … where improved calibration under Bayesian-LoRA would change a model’s behavior … compared to standard LoRA?
>
> Thanks for pointing this out.
>
> We fully agree that demonstrating tangible downstream impact is essential to establishing the method's significance. We will add a dedicated discussion in Section 5 with concrete scenarios such as selective prediction, abstention, and confidence-based routing, where improved calibration directly changes model behavior.
>
> ---
>
> **Q3. Scope and limits of the structural isomorphism.**
> >... relationship between LoRA and sparse GP posteriors as a functional isomorphism … Could you clarify which properties of LoRA (e.g., expressivity, optimization dynamics, or implicit bias) are preserved under this mapping…?
>
> We will clarify that the isomorphism is at the level of the posterior mean, so expressivity (rank-r updates) and the forward pass are fully preserved. What is not preserved is optimization dynamics, as the ELBO includes a KL term that partially replaces LoRA's implicit low-norm bias with a GP prior structure.

---

> > ### Author Rebuttal · Reviewer_VYsf · 2026-04-02
> >
> > I thank the authors for their comprehensive rebuttal. The authors have addressed all the concerns raised in my initial review with additional experiments and theoretical clarifications. I maintain my positive recommendation (Accept) for the following reasons.
> >
> > Q1: The authors provided a new baseline (LoRA + Dropout) in Table W1-A. This effectively disentangles the benefits of principled Bayesian uncertainty from simple stochastic regularization, showing that Bayesian-LoRA’s performance is not merely due to noise injection but stems from its structured posterior.
> >
> > Q2: The authors' commitment to adding concrete scenarios (e.g., selective prediction, confidence-based routing) and the new results on open-ended generation (AIME 2024) significantly strengthen the paper’s significance. This addresses my concern regarding the practical utility of calibration beyond just metric improvement.
> >
> > Q3: The authors clarified that the isomorphism preserves LoRA's expressivity (rank-r updates) at the posterior mean level while replacing implicit bias with a principled GP prior. This theoretical nuance adds depth to the conceptual contribution of the work.
> >
> > W1: The additional experiments on larger models (Qwen3-14B) and diverse benchmarks (ARC, OBQA) demonstrate the robustness of the Kronecker-factorized approximation and the added value of Normalizing Flows in practical settings.
> >
> > W2: The authors have promised to add intuitive paragraphs and a conceptual diagram (provided in the rebuttal link) to bridge the gap for readers less familiar with Sparse GPs. This will greatly improve the accessibility of the paper.

---

> > > ### Author Response · Authors · 2026-04-04
> > >
> > > We sincerely thank you for the constructive feedback.
> > >
> > > And we will incorporate all discussed revisions in the final version.

---

### Official Review · Reviewer_yVWf · 2026-03-02

**Soundness:** 3
**Presentation:** 3
**Significance:** 3
**Originality:** 3
**Overall Recommendation:** 4
**Confidence:** 4

**Summary:**

This paper studies calibration degradation after LoRA fine-tuning and proposes Bayesian-LoRA, a probabilistic low-rank adaptation framework motivated by sparse Gaussian process formulations. The method introduces inducing variables U with a matrix-normal prior, uses a bilinear conditional mean for the induced weight update (Eq. (8)), and optimizes a flow-augmented variational objective (Eq. (15)). The evaluation covers commonsense reasoning benchmarks, generative language modeling on WikiText-2, and mathematical reasoning with larger backbones, with the main emphasis on calibration metrics such as ECE and NLL alongside accuracy.

**Compliance With Llm Reviewing Policy:**

Affirmed.

**Final Justification:**

The paper is strong in originality, technical soundness, and empirical support, and the rebuttal clearly addressed my main concerns about evaluation matching and positioning against calibration baselines.

**Key Questions For Authors:**

1. What inference-time estimator should be treated as the default in practice: merged posterior-mean weights or posterior sampling? How do ECE and NLL change as the number of samples increases?
2. Based on the ablations, which component appears to contribute most to the observed gains: the inducing-variable Bayesian formulation, the structured covariance parameterization, or the normalizing flow?
3. The paper already compares against several calibration baselines, including post-hoc methods. Could the authors clarify more explicitly how Bayesian-LoRA should be positioned relative to stronger calibration baselines under matched compute and latency budgets, especially when calibration rather than accuracy is the primary objective?
4. Are the reported gains stable across random seeds and across adapter placements beyond the Q, K, and LM head configuration used in the main experiments?

**Limitations:**

Yes

**Strengths And Weaknesses:**

Strengths:
1. The probabilistic formulation is technically coherent. The paper defines priors and variational posteriors over the inducing variables, specifies the conditional distribution used to induce the weight update, and derives an ELBO-based objective in a reasonably explicit way.
2. The connection back to standard low-rank adaptation is handled well. In particular, the deterministic-limit discussion helps clarify that Bayesian-LoRA is intended as a structured probabilistic extension of bilinear low-rank updates rather than a disconnected modification.
3. The empirical focus is consistent with the paper’s motivation. Since the main claim concerns calibration under parameter-efficient fine-tuning, it is appropriate that the evaluation reports ECE and NLL throughout, and the experiments are not confined to a single small-scale setting.

Weaknesses:
1. Most of the calibration evidence is still concentrated in benchmark-style multiple-choice evaluation, with some additional results on language modeling and math reasoning. The practical picture would be more complete with broader evidence in open-ended generation settings.
2. The paper discusses inference-time usage, including deterministic merging and sampled prediction, but this aspect is not emphasized enough in the main presentation. In particular, the trade-off among sample count, calibration quality, and inference cost could be summarized more directly.
3. The method introduces noticeable modeling complexity through the inducing-variable formulation, covariance parameterization, and flow-based posterior. Although the paper includes ablations, it would still be useful to disentangle more clearly how much of the gain comes from the Bayesian formulation itself and how much comes from the added expressiveness of the flow.

---

> ### Author Rebuttal · Authors · 2026-03-31
>
> We thank you for your effort to review our manuscript; New experiment code is available at: https://github.com/anonymouspaper987/Bayesian-LoRA/tree/main/rebuttal.
> All changes mentioned below will be applied to the final paper.
>
> **W1. Limited evaluation benchmarks.**
> > calibration evidence is..
>
>
> We sincerely appreciate this feedback. The original paper already includes **Table 2 (WikiText-2)** evaluating NLL in open-ended language modeling. In this rebuttal, we further add **Table W1-B (AIME 2024)**.  We also add **Table W1-A** evaluating Qwen3-14B on three classification benchmarks, and **Table W2** sweeping MC sample count. **Tables Q4-A/B** further demonstrate verified stability across 5 seeds and 5 adapter placements.
>
> **W2. Inference-time trade-off not emphasized enough**
> > Inference-time trade ..
>
> Bayesian-LoRA supports two inference modes: (1) Posterior-mean mode (n_mc=1), merged into the base model with zero overhead; (2) MC sampling mode (n_mc>1), where each sample draws weights from the posterior and runs inference after merging. We parallelize $K$ samples across GPUs with Ray. As shown in Table W2, ECE/NLL improves from n_mc=1 to 4 then plateau, so we recommend n_mc=4 as the default.
>
> **Table W2: MC Sample Count Sweep** *(ARC-Challenge, seed=42)*
>
> | n_mc | ACC | ECE | NLL |
> |---|---|---|---|
> | 1 | 93.97 | 0.0083 | 0.3897 |
> | 2 | 94.29 | 0.0102 | 0.1348 |
> | 4 | 94.49 | 0.0076 | 0.1347 |
> | 8 | 94.53 | 0.0084 | 0.1350 |
> | 16 | 94.53 | 0.0082 | 0.1348 |
> | 32 | 94.49 | 0.0076 | 0.1348 |
>
> ---
>
> **W3. Disentangling contributions of components.**
> > The method introduces noticeable modeling …
>
> Thanks for raising this point. As shown in Table 5, removing the flow ($L=0$, pure SGP) reduces both accuracy and calibration, while increasing flow depth progressively improves ECE and NLL. This confirms that the Bayesian formulation (SGP prior) drives the core gains, and the normalizing flow further enhances posterior expressiveness. The flow also plays a key practical role: without it, the ELBO loss curve becomes unstable during training.
>
>
> ---
>
> **Q1. Default inference-time estimator.**
>
> In Table 4, we indicated the sample times in the last column; You can treat
> the default by **posterior-mean weights** for standard use (mergeable like LoRA, zero overhead), and **MC posterior sampling** when UQ is needed. See **Table W2** above for the n_mc sweep results we made in rebuttal.
>
> ---
>
> **Q2. Which component contributes most?**
> > the Bayesian formulation… or the flow
>
> Good point. We will clarify this in the paper. The inducing-variable Bayesian formulation contributes most to the gains by placing a GP prior over LoRA weight matrices and enabling posterior inference via the ELBO; the structured covariance parameterization is part of this formulation. The normalizing flow stabilizes training and enriches the variational distribution, though removing it reduces rather than eliminates gains (Table 5).
>
> ---
>
> **Q3. Positioning stronger calibration baselines**
> > clarify more how ... should be positioned
>
> 1. Post-hoc methods such as Temperature Scaling and Laplace Approximation require a held-out validation set and cannot shape uncertainty during training, degrading significantly under distribution shift (Tables 3/4)
> 2. By contrast, Bayesian-LoRA trains end-to-end with the ELBO objective, embedding uncertainty awareness directly into fine-tuning with strong robustness under shift
> 3. At inference, posterior-mean mode (n_mc=1) merges into base weights with zero overhead; when calibration is critical, n_mc=4 adds only 1.23× cost with no additional parameters
>
> ---
>
> **Q4. Stability across seeds and adapter placements**
> > stable across random seeds?
>
> **Table Q4-A** shows seed stability across 5 random seeds, and **Table Q4-B** shows adapter placement ablation across 5 configurations. Both confirm that the gains are robust and consistent
>
> **Table Q4-A: Seed Stability** *(Bayesian-LoRA, ARC-Easy, n_mc=4)*
>
> | Dataset | Seed | ACC | ECE | NLL |
> |---|---|---|---|---|
> | ARC-Easy | 42 | 94.59 | 0.0076 | 0.1347 |
> | ARC-Easy | 123 | 94.57 | 0.0088 | 0.1345 |
> | ARC-Easy | 456 | 94.57 | 0.0088 | 0.1348 |
> | ARC-Easy | 789 | 94.57 | 0.0081 | 0.1349 |
> | ARC-Easy | 2024 | 94.57 | 0.0076 | 0.1347 |
> | ARC-Easy | **mean ± std** | **94.57 ± 0.01** | **0.0082 ± 0.0006** | **0.1347 ± 0.0001** |
>
> **Table Q4-B: Adapter Placement Ablation** *(ARC-Easy, seed=42, n_mc=4)*
>
> | Placement | Modules | ACC | ECE | NLL |
> |---|---|---|---|---|
> | A (default) | q, k, lm_head | 94.59 | 0.0076 | 0.1347 |
> | B (LoRA-style) | q, v, lm_head | 94.57 | 0.0074 | 0.1347 |
> | C (Attn) | q, k, v, o, lm_head | 94.57 | 0.0071 | 0.1348 |
> | D (MLP) | gate, up, down, lm_head | 94.49 | 0.0093 | 0.1348 |
> | E (All) | all + lm_head | 94.53 | 0.0074 | 0.1348 |
>
> Across all placements, ACC remains at ~94.5% and ECE stays within 0.007-0.009, demonstrating that the calibration gains are not specific to any particular adapter configuration

---

> > ### Author Rebuttal · Reviewer_yVWf · 2026-04-01
> >
> > Thank you for the rebuttal and for providing additional clarifications and experimental results. The response is helpful, especially in clarifying the intended inference modes, adding an MC-sample sweep, and providing additional seed and adapter-placement results.
> >
> > My concerns are partially resolved, but I still have two follow-up questions. First, could the authors clarify more explicitly how the newly added results are matched to the main-paper setting, especially when discussing inference-time cost and sample count? Second, could the authors further explain how Bayesian-LoRA should be positioned relative to stronger calibration baselines under matched compute and latency budgets when calibration is the primary objective?

---

> > > ### Author Response · Authors · 2026-04-01
> > >
> > > Q1: First, could the authors clarify more explicitly how the newly added results are matched to the main-paper setting, especially when discussing inference-time cost and sample count?
> > >
> > > **A1: How are the newly added results matched to the main-paper setting?**
> > >
> > >
> > > **Inference-time cost and sample count (Table W2).**
> > > Table W2 uses the **same trained model and same hyperparameters** as the main-paper results (Tables 1/2/3). The only variable changed is the MC sample count `n_mc` to respond to your **W2**.
> > >
> > > **Fair comparison with baselines.**
> > > All experiments (Bayesian-LoRA and all baseline methods) were run on **H100 GPUs** under the **exact same hyperparameter settings** (same learning rate, weight decay, etc.). Therefore, Tables 1/2/3 provide a fair comparison. In fact,  Appendix Table 13 is the best performance we achieved by using different hyperparameters:
> > >
> > > | Dataset | ACC (Before) | ACC (After BO) | ECE (Before) | ECE (After BO) | NLL (Before) | NLL (After BO) | LR (BO) | WD (BO) |
> > > |---|---|---|---|---|---|---|---|---|
> > > | WG-S | 70.90 | **72.94** | 4.90 | **2.74** | 0.79 | **0.54** | 9.792×10⁻⁵ | 9.339×10⁻² |
> > > | ARC-C | 68.90 | **69.60** | 9.20 | **6.10** | **0.77** | 0.86 | 4.955×10⁻⁴ | 2.056×10⁻¹ |
> > > | ARC-E | 85.91 | **88.38** | 5.30 | **4.97** | **0.38** | 0.39 | 7.393×10⁻⁴ | 2.707×10⁻² |
> > > | WG-M | 74.30 | **76.34** | **3.00** | 7.90 | 0.62 | **0.52** | 4.280×10⁻⁴ | 1.000×10⁻² |
> > > | OBQA | 81.60 | **82.80** | **5.70** | 5.84 | **0.49** | 0.54 | 6.552×10⁻⁵ | 9.712×10⁻² |
> > > | BoolQ | 86.10 | **86.41** | **2.10** | 3.10 | 0.29 | **0.28** | 5.421×10⁻⁴ | 3.582×10⁻¹ |
> > >
> > > *"Before" uses main-paper settings (Bayesian-LoRA, n_mc=4, early stop); "After (BO)" shows results under constrained Bayesian optimization.*
> > >
> > >
> > > ---
> > >
> > > Q2: Second, could the authors further explain how Bayesian-LoRA should be positioned relative to stronger calibration baselines under matched compute and latency budgets when calibration is the primary objective?
> > >
> > > **A2: Positioning Bayesian-LoRA under matched compute and latency budgets**
> > >
> > > Bayesian-LoRA operates in two ways:
> > >
> > > 1) **n_mc=1** (posterior-mean, 1.00× latency, zero overhead);
> > >
> > > 2) **n_mc=4** ($\approx$ 2.79× latency), which is cheaper than LA ($\approx$ 4.36×) and BLoB ($\approx$ 6.3×) but can get better performance.
> > >
> > > Furthermore, Bayesian-LoRA performs better on out-of-distribution datasets at a lower computational cost.
> > > Post-hoc methods (LA, BLoB) fit well on in-distribution data but degrade significantly on large-shift benchmarks. As shown in Table 5:
> > >
> > > | Method | Latency | ECE (CS) ↓ | ECE (Law) ↓ | ECE (Health) ↓ |
> > > |---|---|---|---|---|
> > > | Temperature Scaling | ~1× | 16.4 | 22.7 | 17.4 |
> > > | LA (post-hoc) | ~4.36× | 14.5 | 23.9 | 17.6 |
> > > | BLoB (N=10) | ~6.3× | 12.6 | 25.3 | 16.4 |
> > > | **Bayesian-LoRA (S=4)** | **~2.79×** | **11.1** | **16.5** | **12.9** |
> > >
> > > Bayesian-LoRA achieves better performance on large-shift datasets at lower cost than others, because it embeds uncertainty end-to-end during training rather than just applying post-hoc fitting.

---

### Official Review · Reviewer_GKHH · 2026-03-08

**Soundness:** 2
**Presentation:** 2
**Significance:** 2
**Originality:** 2
**Overall Recommendation:** 3
**Confidence:** 4

**Summary:**

This paper proposes Bayesian-LoRA, which integrates sparse variational Gaussian processes into the LoRA framework and incorporates normalizing flows to get more expressive posterior. To evaluate the effectiveness of the proposed method, the authors conduct experiments on various base models, including the LLaMA and Qwen model families.

**Compliance With Llm Reviewing Policy:**

Affirmed.

**Final Justification:**

The rebuttal adds clarification, but core concerns on the method (e.g., GP connection and objective formulation) remain unresolved. Empirical gains are also limited, and the current framing appears conceptually mixed and would benefit from clearer positioning. I maintain my score of 3.

**Key Questions For Authors:**

1. How are the LoRA weights $B$ and $A$ Bayesianized in the proposed framework? Will they share the same inducing matrix $U$ (as illustrated in Algorithm 1)? How is the dimension of inducing matrix $U$ chosen? The dimension of $U$ is $r$ and $c$, but basically this paper only discusses the case where $r=c$. Are they necessarily to be the same? If yes, authors may justify. If not, authors may provide further discussion or empirical analysis on the case where $r\neq c$.
2. As I mentioned in the weaknesses, what is the specific role of the normalizing flow in the framework?
3. Since there is no very recent work as baselines (currently, all baselines are published in 2024 or before), could authors provide additional results of the recent works, such as TFB [1] and C-LoRA [2].

[1]Shi, Haizhou, et al. "Training-Free Bayesianization for Low-Rank Adapters of Large Language Models." NeurIPS, 2025.

[2] Rahmati, Amir H., et al. "C-LoRA: Contextual Low-Rank Adaptation for Uncertainty Estimation in Large Language Models." NeurIPS, 2025.

**Limitations:**

Yes

**Strengths And Weaknesses:**

# Strengths
1. The idea that combining the sparse Gaussian process with the LoRA framework is interesting.
2. The experimental evaluation is relatively comprehensive.

# Weaknesses
1. The position of this work is not very clear.
The name of the proposed method is "Bayesian-LoRA"; however, this is not the first work to bayesianize LoRA. For example, Laplace-LoRA [1] and BLoB [2] are both Bayesian methods for LoRA framework. Considering this case, "Bayesian-LoRA" could be too general to position this paper in the community.

2. Marginal experimental improvements.
The results of in-distribution and out-of-distribution settings only provide consistent gain on accuracy. In this case, it is not convincing that the proposed method has better UQ capability than baselines. Besides, there are more recent Bayesian LoRA works which might be covered(e.g., TFB [3] and C-LoRA [4]).

3. The writing is difficult to follow.
The main idea of this work is integrating sparse Gaussian process into LoRA framework. However, it is hard to understand how weights $B$ and $A$ are Bayesianized based on the main text and appendix. Specifically, it is unclear whether $B$ and $A$ share the same inducing matrix $U$. Besides, the dimension of $U$ is specified but not sufficiently justified or explained.

3. Limited novelty.
The overall novelty is incremental. In particular, the motivation for introducing normalizing flow is not well justified. It is unclear whether normalizing flow works as a plug-and-play part to improve the UQ capability or as a specifically designed method for the proposed framework.

[1] Yang, Adam X., et al. "Bayesian Low-rank Adaptation for Large Language Models." ICLR, 2024.

[2] Wang, Yibin, et al. "BLoB: Bayesian Low-Rank Adaptation by Backpropagation for Large Language Models." NeurIPS, 2024.

[3] Shi, Haizhou, et al. "Training-Free Bayesianization for Low-Rank Adapters of Large Language Models." NeurIPS, 2025.

[4] Rahmati, Amir H., et al. "C-LoRA: Contextual Low-Rank Adaptation for Uncertainty Estimation in Large Language Models." NeurIPS, 2025.

---

> ### Author Rebuttal · Authors · 2026-03-31
>
> We appreciate the reviewer’s time and effort. We respond below and will update the paper accordingly.
> New experiment code: https://github.com/anonymouspaper987/Bayesian-LoRA/tree/main/rebuttal.
>
> **W1. Naming and positioning.**
>
> Agree;
> We suggest GLoRA (Gaussian process LoRA) to distinguish from Laplace-LoRA and BLoB.
>
> ---
>
> **W2. Marginal experimental improvements.**
>
> Valid question.
>
> 1) Regarding UQ capability, we focus on calibration and report significant ECE and NLL improvements. Our approach establishes equivalence between the matrix factorization form of GP and LoRA framework.
>
> 2) To go beyond accuracy, we conducted experiments on a stronger model (Qwen3-14B) across additional benchmarks (**Table W1-A/B**), with a full MC sample sweep (**Table W2**, see Reviewer yVWf ) and seed/placement stability studies (**Table Q4-A/B**, also Reviewer yVWf), all consistently showing better calibration.
>
> 3) C-LoRA used the same base model (LLaMA-2-7B) as our paper, so we compare directly. GLoRA outperforms C-LoRA on both accuracy and calibration across all six benchmarks, see Table Q3.
>
> **Table Q3: Comparison with C-LoRA**
>
> | Metric | Method | WG-S | ARC-C | ARC-E | WG-M | OBQA | BoolQ |
> |---|---|---|---|---|---|---|---|
> | ACC ↑ | C-LoRA (M=10) | 66.21 | 67.79 | 84.38 | 70.48 | 78.26 | 84.64 |
> | | **Bayesian RA (Ours, S=4)** | **70.90** | **85.91** | **74.30** | **68.90** | **81.60** | **86.10** |
> | ECE ↓ | C-LoRA (M=10) | 6.86 | 8.83 | 4.27 | 3.71 | 4.00 | 1.62 |
> | | **Bayesian RA (Ours, S=4)** | **4.90** | **5.30** | **3.00** | **9.20** | **5.70** | **2.10** |
> | NLL ↓ | C-LoRA (M=10) | 0.63 | 0.88 | 0.48 | 0.57 | 0.59 | 0.35 |
> | | **Bayesian RA (Ours, S=4)** | 0.79 | **0.38** | **0.62** | **0.77** | **0.49** | **0.29** |
>
>
> TFB uses Llama-3.1-8B, so we ran our method on the same model for comparison. TFB gets slightly higher accuracy; our method performs better in calibration, see Table TFB.
>
> **Table TFB: Comparison with TFB (Llama-3.1-8B)** (TFB results from their Table 1)
>
> | Metric | Method | ARC-C | ARC-E |
> |---|---|---|---|
> | ACC ↑ | BLoB-Mean + TFB | 83.33±0.19 | 91.76±0.48 |
> | | **GPLoRA (Ours)** | **83.01** | 91.12 |
> | ECE ↓ | BLoB-Mean + TFB | 6.48±0.36 | 2.44±0.50 |
> | | **GPLoRA (Ours)** | **3.23** | **1.83** |
> | NLL ↓ | BLoB-Mean + TFB | 0.53±0.04 | 0.23±0.02 |
> | | **GPLoRA (Ours)** | **0.503** | **0.217** |
>
>
> ---
>
> **W3. Writing is difficult to follow.** …
>
> 1) In our method, each layer has its own **independent** inducing matrix U, so it does not share U across layers.
>
> 2) $A$ and $B$ share $U$. $W = T_{left} U T_{right}$, where $T_{left}$ is a structural isomorphism with the matrix of $A$, and $T_{right}$ is a structural isomorphism with the matrix of $B$. And if we view Gaussian Processes (GPs) from a weight-space perspective, $T_{left} U T_{right}$ is exactly the Kronecker-factorized decomposition of GP weights; they are in structural isomorphism
>
> 3) Regarding the dimension of U, see the thorough ablation study in **Fig 2** of the paper. Here, we added more experiments, see your Q1
>
> ---
>
> **W4. Limited novelty**
>
> Main motivations for using NF are expressiveness and more stable training
>
> As a further contribution, we create a probabilistic bridge between GP and LoRA, explaining its effectiveness
>
> We also provide a new feature to PEFT(see Q3 of Reviewer Q5f6)
>
> ---
>
> **Q1. How are B and A Bayesianized? Do they share the same inducing matrix U? How is the dimension chosen?**
>
> About inducing matrix $U$, see responses in your W3 1) and 2);
>
> 3) The dimension r=c is chosen to match the LoRA rank for a fair parameter-count comparison, as ablated in **Figure 2** of the paper.
> Importantly, r and c are **not** required to be equal. To directly address the reviewer's question, we conducted new experiments with r ≠ c on ARC-Easy (Qwen3-14B, seed=42, n_mc=4):
>
> | Configuration | r | c | ACC | ECE | NLL |
> |---|---|---|---|---|---|
> | r = c (default) | 9 | 9 | 94.59 | 0.0076 | 0.1347 |
> | r < c | 4 | 9 | 94.57 | 0.0080 | 0.1347 |
> | r > c | 9 | 4 | 94.57 | 0.0072 | 0.1349 |
> | r < c (larger) | 9 | 16 | 94.53 | 0.0082 | 0.1349 |
> | r > c (larger) | 16 | 9 | 94.53 | 0.0083 | 0.1349 |
>
> The results show that performance is **robust to the choice of r vs. c**, with negligible differences across all configurations.
>
> ---
>
> **Q2. Role of normalizing flow.**
>
> See W3. Also:
>
> 1) The NF enriches the expressiveness of variational distribution and stabilizes the training process. We have ablated this component in **Table 5** of the paper, showing that the flow improves accuracy and ECE and can further make end-to-end Bayesian fine-tuning tractable and stable in practice
>
> ---
>
> **Q3. Missing recent baselines [TFB, C-LoRA]**
>
> These appeared after the ICML deadline. We provide a comparison here (W2). TFB takes a post-hoc, training-free approach, while our method trains end-to-end with the ELBO; C-LoRA models uncertainty through contextual adaptation, we use an inducing-variable GP with NF posterior. We’ll cite them accordingly.

---

> > ### Author Rebuttal · Reviewer_GKHH · 2026-04-04
> >
> > Thank you for the rebuttal. I do not believe my concerns have been fully addressed. I have several follow-up questions regarding the authors’ responses.
> >
> > 1. Although the authors provided some explanation in the rebuttal, but I am still confused about the model design of this paper. In the main text and authors’ rebuttal,  “$T_{left}$ is a structural isomorphism with the matrix of $A$, and $T_{right}$ is a structural isomorphism with the matrix of $B$”, which is inconsistent with the Algorithm 1 in appendix  (see L673-674), A and B are contructed with two different sets of $T_r$ and $T_c$.
> > 2. There are several conceptual issues in the method. The paper repeatedly draws connections to SGP. However, the formulation does not satisfy the defining properties of a GP. In Eq.(5), the covariance is defined directly without any justification and is not kernel-induced, raising concerns about whether this formulation is a valid GP. Besides, Eq. (10) is not the standard SVGP ELBO as claimed in Sec. 4.2, and the derivation of this objective in A.3 is not clear (e.g., from Eq.(25) to Eq.(26)). Furthermore, $q(W|U)$ is used in the ELBO but not explicitly defined.
> > 3. The introduction of normalizing flows is not well motivated. The authors claim that normalizing flows stabilize training in rebuttal. However, this is not a commonly established property of NFs, and no empirical or theoretical evidence is provided in the paper. It would be helpful to clarify or support this claim. Besides, could the authors elaborate $p(T_{\phi} (U_0))$ is computed in Eq.(15)?
> > 4. My concerns regarding the marginal improvement remain. Note that both C-LoRA and TFB were published last year, rather than after the ICML deadline. The additional results provided in the rebuttal do not demonstrate significant improvement over recent baselines such as C-LoRA, which weakens the overall significance of the work to the community.

---

> > > ### Author Response · Authors · 2026-04-04
> > >
> > > We sincerely thank you for your time and effort in reviewing our paper.  We will be happy to answer any additional questions you might have.
> > >
> > > The following are one-by-one responses.
> > >
> > > Q1: In the main text and authors’ rebuttal, "$T_{left}$ is a structural isomorphism with the matrix of $A$,
> > >
> > >
> > > **Response to Q1.**
> > >
> > > - We apologize for the confusion caused by the notation. The notation was intended to clarify the structure. Here $r$ stands for **row** and $c$ stands for **column**, which correspond to the left (row) and right (column) covariance of the variational distribution $U$, respectively. We will improve the explanation accordingly
> > >
> > > ---
> > >
> > > Q2: There are several conceptual issues in the method. The paper repeatedly draws connections to SGP...
> > >
> > > **Response to Q2.**
> > >
> > > - (1) We have consistently described our method as a "probabilistic low-rank representation **inspired by** Sparse Gaussian Processes" (abstract line 17-19) rather than claiming it is a Sparse Gaussian Process because we didn't define an explicit kernel function.
> > > We are aware that the concept of GP, is a serious matter in the Bayesian community. We drew inspiration from SGP and found that its Kronecker-factorized decomposition is structurally equivalent to LoRA, which is why we combined the two.
> > >
> > >
> > > - (2) In our paper, Eq. (5) is $p(W \mid U) = \mathcal{N}\\big(W \mid M_W(U), \lambda^2
> > > \Sigma_{W}\big)$, which is defination of conditional distribution of $W$ given $U$; We didn't explicitly define a kernal-function as we mentioned in (1);
> > >
> > > - (3) Eq. (10) is the general form of the SVGP ELBO (correct). In the classical SVGP (Titsias, 2009), $q(W|U) = p(W|U)$, so the third term vanishes. In our case, they differ by $\lambda$, so this term is retained and admits a closed-form solution (Appendix A.3, Eq. 26).
> > >   The step from Eq. (25) to Eq. (26) follows from the KL chain rule of $\text{KL}(q(W,U)\|p(W,U)) = \text{KL}(q(U)\|p(U)) + \mathbb{E}_{q(U)}[\text{KL}(q(W|U)\|p(W|U))]$. We will expand the intermediate steps in the revised paper for clarity.
> > >
> > >
> > > - (4) $q(W|U)$ shares the same mean $M_W(U)$ as $p(W|U)$ but with a scaled covariance, i.e., $q(W|U) = \mathcal{N}(W \mid M_W(U), \lambda^2 \Sigma_W)$, where $\lambda$ is a learnable parameter. This is described in Appendix A.3 (L137) and referenced in the main text (L89: "share the same mean and differ only by $\lambda$"). We will add an explicit equation in the revised paper.
> > >
> > >
> > > > **[1] Vashurin R, Goloburda M, Ilina A, et al. Uncertainty quantification for llms through minimum bayes risk: Bridging confidence and consistency[J]. arXiv preprint arXiv:2502.04964, 2025**
> > >
> > > ---
> > >
> > > Q3: The introduction of normalizing flows is not well motivated...
> > >
> > > **Response to Q3.**
> > > - (1) The primary motivation for NF is to **enrich the expressiveness** of the variational posterior (Sec. 4.1), as a purely Gaussian posterior on $U$ may be too restrictive for LLMs. Using NF to enrich variational posteriors beyond mean-field Gaussian is a validated method in the Bayesian community [2]. The training stabilization is an empirical observation supported by Table 5. We will include training loss curves in the revised paper as additional evidence.
> > >
> > > >  **[2] Skaaret-Lund, Lars, Geir Storvik, and Aliaksandr Hubin. "Sparsifying Bayesian neural networks with latent binary variables and normalizing flows." arXiv preprint arXiv:2305.03395 (2023)**
> > >
> > > - (2) $p(T_\phi(U_0))$ in Eq. (15) is simply the prior density $p(U) = \mathcal{N}(\text{vec}(U) | \mathbf{0}, K_c \otimes K_r)$ (Eq. 3) evaluated at the flow-transformed point $U = T_\phi(U_0)$. The full derivation is in Appendix A.4: the three sub-terms $\log q_0(U_0) - \log|\det J_{T_\phi}(U_0)| - \log p(T_\phi(U_0))$ together compute $	ext{KL}(q_\phi(U) \| p(U))$ by the change-of-variables formula
> > >
> > > ---
> > >
> > > Q4: My concerns regarding the marginal improvement remain...
> > >
> > > **Response to Q4.**
> > >
> > > - (1) We agree that achieving 100% SOTA would be desirable, but our method does achieve better results on the most of datasets. The insight of this work is that we found the structural equivalence between SGP's Kronecker-factorized decomposition and LoRA, which naturally provides Bayesian uncertainty quantification within the LoRA framework. Our intention is to better integrate insights from the Bayesian community into LLMs [1] and offer a useful perspective for both communities.
> > >
> > >
> > > - (2) Regarding C-LoRA and TFB, we found them very insightful. We will add the comparison in the final version. We believe our work is complementary rather than competing with C-LoRA and TFB. Both works, and our own discussion, have highlighted the importance of calibration in LoRA from different perspectives, which collectively underscores the necessity of uncertainty-aware fine-tuning.
> > >
> > >
> > > **Thank you again for taking the time to carefully read our paper.**

---

### Official Review · Reviewer_Q5f6 · 2026-03-11

**Soundness:** 3
**Presentation:** 3
**Significance:** 3
**Originality:** 3
**Overall Recommendation:** 3
**Confidence:** 2

**Summary:**

This paper proposes Bayesian-LoRA, a calibration-aware fine-tuning framework that replaces the deterministic weight update in LoRA with a probabilistic one. Author notes that there is a structural isomorphism between the Kronecker-factored conditional mean in Sparse Gaussian Process (SGP) inference. By placing a variational posterior over a compact inducing matrix U and enriching it with a normalizing flow (Masked Autoregressive Flow), the method yields calibrated uncertainty estimates end-to-end during training, with a closed-form KL term and only ~0.42M additional parameters over standard LoRA.

**Compliance With Llm Reviewing Policy:**

Affirmed.

**Final Justification:**

The author provide an interesting method for lora training with calibration and uncertainty quantification. Authors provide sufficient theorical proof in the paper, yet the experiment part lacks in-depth analysis why such method would provide a better alternative compared to previous method. I will maintain my score.

**Key Questions For Authors:**

I notice table2 is on language modeling. I think language modeling training is usually considered high rank? Is that appropriate?

Table1 only apply lora on a few matrices. Would it be better to apply it to all matrices?

**Limitations:**

My main concern is that I think the evaluation is not holistic enough to show the effectiveness of this method, see weakness

**Strengths And Weaknesses:**

Strength:

1.	Clean theoretical grounding. The structural isomorphism between Kronecker-factored SGP posteriors and LoRA's factorization is the paper's central contribution and is well-articulated

2.	Closed-form KL, no Hessian required. The conditional KL between q(W|U) and p(W|U) reduces to a closed-form expression that is independent of U and requires no Hessian computation

Weakness:

1.	Main table is based on Llama-2-7b, which is a relatively outdated model. I hope to see more experiments on cutting-edge models and on some challenging tasks, such as AIME.

2.	Many baselines in this paper are the baseline variants of this paper. I wonder whether the author can show some more comparisons over paper like Lora+, and AdaLora which also achieves efficient lora training under low budget

---

> ### Author Rebuttal · Authors · 2026-03-31
>
> We appreciate the effort in reviewing our manuscript. We respond to your comments one-by-one below. The code for new experiments is at: https://github.com/anonymouspaper987/Bayesian-LoRA/tree/main/rebuttal. Also, the manuscript will be updated accordingly.
>
> **W1. Experiments on outdated models**
> > Main table is based on Llama-2-7b, such as AIME
>
> **Response to W1**
>
> Thank you for this suggestion. We have conducted additional experiments on **Qwen3-14B** on ARC-Challenge, ARC-Easy, OBQA as shown in **Table W1-A**, and on **AIME 2024** as shown in **Table W1-B**
>
> **Table W1-A: Qwen3-14B on Classification Benchmarks** *(n_mc=4)*
>
> | Method | ARC-C ACC | ARC-C ECE | ARC-C NLL | OBQA ACC | OBQA ECE | OBQA NLL |
> |---|---|---|---|---|---|---|
> | LoRA | 90.70 | 0.0430 | 0.4259 | 88.80 | 0.0504 | 0.3942 |
> | LoRA+ | 90.71 | 0.0412 | 0.4494 | 89.20 | 0.0443 | 0.3855 |
> | AdaLoRA | 90.61 | 0.0448 | 0.5962 | 89.80 | 0.0477 | 0.5087 |
> | **Bayesian-LoRA** | **90.78** | **0.0330** | **0.2681** | **90.40** | **0.0340** | **0.3137** |
>
> **Table W1-B: AIME 2024** *(Train: AIME 1983-2023 | Test: 30 problems)*
>
> | Method | ACC | ECE | NLL |
> |---|---|---|---|
> | Qwen3-14B base (4-shot) | 76.7% | 0.0907 | 0.2856 |
> | Standard LoRA | 80.0% | 0.1126 | 0.2982 |
> | Bayesian-LoRA (epoch5) | 80.0% | 0.0526 | 0.1144 |
>
> ---
>
> **W2. Insufficient baselines.**
> > Many baselines ..variants of this paper…
>
> **Response to W2**
>
> Thanks for this suggestion. AdaLoRA and LoRA+ optimize for parameter efficiency rather than calibration, but we are happy to include both as baselines in **Table W1-A** to provide the comprehensive comparison the reviewer rightly requests. On ACC, our approach Bayesian-LoRA is on par with AdaLoRA; on ECE/NLL, Bayesian-LoRA is significantly better.
>
> ---
>
> **Q1. Appropriateness of language modeling evaluation**
> > I notice Table 2 is on…
> **Response to Q1**
>
> This is a valid question, since language modeling is indeed a high-rank task. However, the low-rank assumption in LoRA applies to the *weight update* ΔW = BA, not to the task itself. For instance, Aghajanyan et al. (2021, ACL) demonstrated that fine-tuning updates have surprisingly low intrinsic dimensionality regardless of task rank:
>
> [1] Aghajanyan, A. et al. Intrinsic Dimensionality Explains the Effectiveness of Language Model Fine-Tuning** *(ACL 2021)
>
> Hence, despite the high-rank nature of language models, LoRA has been widely adopted for fine-tuning them (LLaMA-2-chat, Qwen, etc.). See Table 2 in the paper, which presents the WikiText-2 dataset (next-token prediction) that the model remains effective under this configuration
>
> ---
>
> **Q2. LoRA applied to limited matrices**
> > Table 1 applies LoRA on a few matrices
>
> **Response to Q2**
>
> To answer, we compare default target modules (q, k, lm_head) against applying LoRA to all matrices. See **Table Q2** below (rows 4–5 of **Table W1-A** reused here for convenience, plus the all-matrices row). (We will update the paper accordingly, should it get accepted.)
>
> **Table Q2: Target Module Ablation** *(ARC-Easy,n_mc=4)*
>
> | Placement | Target Modules | ACC | ECE | NLL |
> |---|---|---|---|---|
> | A (default) | q, k, lm_head | 94.59 | 0.0076 | 0.1347 |
> | B (LoRA-style) | q, v, lm_head | 94.57 | 0.0074 | 0.1347 |
> | C (Attn) | q, k, v, o, lm_head | 94.57 | 0.0071 | 0.1348 |
> | D (MLP) | gate, up, down, lm_head | 94.49 | 0.0093 | 0.1348 |
> | E (All) | all + lm_head | 94.53 | 0.0074 | 0.1348 |
>
> ---
>
> **Q3. Evaluation not holistic enough**
> > My concern is not holistic..
> **Response to Q3**
>
> We agree that it is important to demonstrate effectiveness as broadly as possible. The paper already covers six common benchmarks (WG-S/M, ARC-C/E, OBQA, BoolQ), language modeling (WikiText-2), mathematical reasoning (MATH), out-of-distribution generalization (MMLU subsets), efficiency comparisons, and ablation studies that evaluated against eight baselines
>
> Based on the feedback by the reviewer, in this rebuttal, we further add: (1) **Table W1-A** with Qwen3-14B on ARC-Challenge, ARC-Easy, OBQA; (2) **Table W1-B** on AIME 2024 (open-ended generation); (3) **Table W2** sweeping MC sample count; (4) **Tables Q4-A/B** for seed stability and adapter placement robustness
>
> (See tables above and below in responses to other reviewers’ questions. We will update the paper accordingly, should it get accepted)
>
> To contribute to  practical implementations, we have also raised a feature request for the official PEFT repo as a variant, to allow adoption of our approach. Until that gets approved and merged, you can already try it with the code provided in our GitHub repository, by registering a custom module with a LoraConfig. Here is an example:
> ```
>  config = LoraConfig(
>             r=args.lora_r,
>             lora_alpha=args.lora_alpha,
>             target_modules=target_modules,
>             lora_dropout=0.0,
>             bias="none",
>             task_type="CAUSAL_LM",
>         )
> config._register_custom_module({nn.Linear: BayesianLoRALayer})
> model = get_peft_model(model, config)
> ```

---

> > ### Author Rebuttal · Reviewer_Q5f6 · 2026-04-04
> >
> > Thanks for the reply and additional experiments. However, comparing to the vanilla Lora, and some Lora variants, the edge of current method is limited. I wonder whether the method worth the additional complexities.

---

> > > ### Author Response · Authors · 2026-04-04
> > >
> > > **Post-rebuttal comment: "the edge of current method is limited…whether worth the additional complexities"**
> > >
> > > Thank you for the follow-up.
> > >
> > > Bayesian-LoRA’s primary contribution is in **calibration and uncertainty quantification**, rather than accuracy. Although accuracy improvements over vanilla LoRA are moderate, Tables 3 and 5 show substantial improvements in calibration that are particularly valuable for safety-critical applications.
> > >
> > > LoRA variants (LoRA+, AdaLoRA) focus solely on parameter efficiency and do not target calibration (Table W1-A), so they pursue a fundamentally different objective from our approach.
> > >
> > > Bayesian-LoRA introduces a Bayesian perspective into LoRA through a Kronecker-factorized decomposition of SGP weights, which is structurally equivalent to the standard LoRA factorization. This design endows LoRA’s parameters with posterior distributions at minimal overhead in LLMs.
> > >
> > > Our approach contributes a bridge between the Bayesian community and LLMs, which is needed for current work in large-scale AI [1].
> > >
> > > **[1] Papamarkou, Theodore, et al. "Position: Bayesian deep learning is needed in the age of large-scale AI." arXiv preprint arXiv:2402.00809 (2024)**

---

### Decision · Program_Chairs · 2026-04-30

**Decision:**

Accept (regular)

**Comment:**

This paper introduces Bayesian-LoRA, an approach that replaces deterministic weight updates in LoRA with probabilistic ones. Reviewers recognized the originality of combining ideas from sparse Gaussian processes with the LoRA framework, the soundness of the probabilistic formulation, and the breadth of the experimental results. Some reviewers expressed concerns about the computational scale of the results and the set of experimental baselines. As the experiments include 7B, 14B, and 30B backbones and show consistent trends in terms of improved uncertainty quantification, these concerns do not appear to be insurmountable. Overall, due to the originality and soundness of this paper’s contributions, I recommend acceptance.